# A tri-ionic anchor mechanism drives Ube2N-specific recruitment and K63-chain ubiquitination in TRIM ligases

Leo Kiss[1,4], Jingwei Zeng [1,4], Claire F. Dickson[1,2], Donna L. Mallery[1], Ji-Chun Yang [1], Stephen H. McLaughlin [1], Andreas Boland[1,3], David Neuhaus[1,5] & Leo C. James[1,5]*

The cytosolic antibody receptor TRIM21 possesses unique ubiquitination activity that drives broad-spectrum anti-pathogen targeting and underpins the protein depletion technology Trim-Away. This activity is dependent on formation of self-anchored, K63-linked ubiquitin chains by the heterodimeric E2 enzyme Ube2N/Ube2V2. Here we reveal how TRIM21 facilitates ubiquitin transfer and differentiates this E2 from other closely related enzymes. A tri-ionic motif provides optimally distributed anchor points that allow TRIM21 to wrap an Ube2N~Ub complex around its RING domain, locking the closed conformation and promoting ubiquitin discharge. Mutation of these anchor points inhibits ubiquitination with Ube2N/ Ube2V2, viral neutralization and immune signalling. We show that the same mechanism is employed by the anti-HIV restriction factor TRIM5 and identify spatially conserved ionic anchor points in other Ube2N-recruiting RING E3s. The tri-ionic motif is exclusively required for Ube2N but not Ube2D1 activity and provides a generic E2-specific catalysis mechanism for RING E3s.

[1] Medical Research Council Laboratory of Molecular Biology, Cambridge, UK. [2] Present address: University of New South Wales, Sydney, NSW, Australia. [3] Present address: Department of Molecular Biology, Science III, University of Geneva, Geneva, Switzerland. [4] These authors contributed equally: Leo Kiss, Jingwei Zeng. [5] These authors jointly supervised: David Neuhaus, Leo C. James.  *email: lcj@mrc-lmb.cam.ac.uk

TRIM21 and TRIM5 belong to the largest family of RING E3s, the tripartite motif or TRIM ligases[1]. Both E3s possess an unusual catalytic mechanism whereby they mediate target degradation by transferring ubiquitin to themselves rather than to their substrate[2,3]. TRIM21 is further distinct in that it does not engage its target directly but indirectly using antibodies, which behaves as a natural PROTAC (proteolysis targeting chimera)[4]. These properties of indirect targeting and ubiquitination allow TRIM21 to inhibit infection by a diverse array of pathogens, including viruses, bacteria, and prions, by causing their proteasomal degradation and activating immune signaling[4–7]. Moreover, they allow TRIM21 to deplete endogenous cellular proteins during application of Trim-Away, a technology akin to siRNA but which operates directly at the protein level, is faster and requires no prior cellular modification or DNA transfection[6,7].

The general mechanisms of E2:E3 catalysis are well established. RING E3s typically dimerize and use one RING protomer to recruit an E2, while the second RING protomer engages its charged ubiquitin[8,9]. In so far as RINGs have an active site, this is thought to be provided by a catalytic arginine or lysine residue, called the linchpin, which interacts close to where the C-terminus of ubiquitin is conjugated to the E2 and forms contacts with both proteins[8–10]. Mutation of this linchpin typically reduces, although crucially does not always abolish, catalytic activity. Despite identification of these general principles, key questions of E2:E3 catalysis remain unanswered. For instance, RINGs are typically constitutively active in vitro, but not in cells. How regulation is achieved is poorly understood, although higher-order oligomerization has been linked to the activation of the antiretroviral E3 ligase TRIM5[11]. A second key question is how RING E3s catalyze the formation of specific ubiquitin chain linkages. Chain specificity is determined by the recruited E2[12], but E2s are structurally highly related, and despite numerous studies, it remains unclear how RINGs recruit a specific E2 in order to catalyze synthesis of the required chain type.

In addition to its unusual properties of indirect substrate engagement and self-ubiquitination-driven target degradation, TRIM21 does not conform to many of the general rules of E2:E3s given above. It retains activity as a monomeric RING and uses a domain uniquely found in TRIMs, the B-box, to prevent constitutive ubiquitination by blocking E2 recruitment[13]. Upon its activation during engagement of antibody-coated pathogens or proteins in the cytosol, TRIM21 sequentially recruits the E2-conjugating enzymes Ube2W and Ube2N/Ube2V2[2,5]. It uses Ube2W to modify its own N-terminus with a mono-ubiquitin[2], which is then extended into a K63-linked anchored polyubiquitin chain in conjunction with Ube2N/V2[5]. The detected antibody complex, along with TRIM21 itself, is subsequently degraded by recruitment of the AAA ATPase VCP/p97 and the 26S proteasome[4,14]. During proteasomal processing, the 19S deubiquitinase Poh1 releases K63-chains en bloc, resulting in robust immune signaling via NF-κB, AP-1, and IRF3/5/7[2,5].

In this work, we set out to unravel how TRIM21 recruits its canonical E2 enzyme Ube2N and catalyzes ubiquitin transfer, by utilizing a combination of structural, biochemical, and cell biological approaches. We describe the cellular E2-specific recruitment mechanism in RING E3s and show how it drives ubiquitination both in vitro and in cells. X-ray crystallography and NMR spectroscopy reveal a tri-ionic motif that captures E2~Ub complexes in the active closed state by providing conformation-specific anchor points. We show that this tri-ionic motif is conserved across TRIM RING ligases, and is also used by the antiretroviral TRIM5 to drive K63-ubiquitin chain synthesis. Finally, we identify structurally conserved charged anchor points across divergent RINGs, suggesting that this may be a common mechanism for Ube2N-specific catalysis.

## Results

**TRIM21 catalyzes ubiquitination with redundant E2s in vitro.** Antibody-dependent virus neutralization by TRIM21 is dependent on interaction with its canonical E2 enzymes Ube2W and Ube2N(Ubc13)/Ube2V2(Uev2)[2,5]. However, whether additional E2 enzymes are also required is unclear. For instance, it has been suggested that an additional E2 might be required in order to build K48-chains, which are classically associated with proteasomal recruitment[2]. We therefore performed a biochemical screen for catalytic activity against 26 E2 enzymes (Fig. 1a; Supplementary Fig. 1). Apart from Ube2W and Ube2N, this screen revealed TRIM21 ubiquitination activity with Ube2D1 to Ube2D4 (UbcH5a/b/c/d), Ube2E1 (UbcH6), and Ube2E3 (UbcH9). In contrast to a previous report[15], activation of Ube2E1 and Ube2E3 by TRIM21 was minimal, and ubiquitin was conjugated to the E2 enzymes rather than TRIM21 (Supplementary Fig. 1). The Ube2D E2s are known to catalyze K48-chain formation, but are also highly promiscuous. To analyze their interaction with TRIM21 in more detail, an NMR titration of $^{15}$N-labeled TRIM21-RING$^{M10E}$ (T21-R$^{M10E}$) with Ube2D1 was performed (Fig. 1b). This TRIM21 construct greatly improved spectral quality relative to WT, since it suppresses the weak dimerization of the WT RING domain, while retaining ubiquitination activity[13]. The titration revealed that Ube2D1 interacts with TRIM21 in a comparable manner to Ube2N (Fig. 1c). Ube2D enzymes are widely used in E3 ubiquitination assays, but evidence for their involvement in specific physiological functions is scarce. We investigated whether Ube2D is required for TRIM21-dependent virus neutralization. In Ube2D1/Ube2D2/Ube2D3-depleted cells, antibody-dependent virus neutralization was indistinguishable from WT, as opposed to the almost complete lack of neutralization in TRIM21-depleted cells (Fig. 1d–f). Importantly, this suggests that TRIM21 must be capable of differentiating between E2s, and that this property is not captured when using in vitro assays.

**Crystal structure of TRIM21-RING with Ube2N~Ub.** Our attempt to unpick the basis of fine E2 specificity began with solving the complex between T21-R and a stable isopeptide-linked Ube2N~ubiquitin (Ube2N~Ub) conjugate (Supplementary Table 1 and Supplementary Fig. 2). The structure was solved to 2.8 Å resolution with two full complexes (2xT21-R and 2xUbe2N~Ub) in the asymmetric unit. The overall domain orientation shows Ube2N~Ub in the closed conformation primed for catalysis bound by the two protomers in the RING dimer (Fig. 2a, b). While Ube2N only contacts the proximal RING domain, ubiquitin is bound by both the proximal and distal RING. Typically, E2s interact with the RING via residues in helix 1 and loops 4 and 7[12]. At the E2:E3 interface, there are a number of hydrophobic interactions, mediated by aromatic side chains, and conserved hydrogen bonds. The generic interactions present in our structure include the Ube2N S96 sidechain, which forms a hydrogen bond to TRIM21 P52 carbonyl, and Ube2N R7 sidechain, which forms a hydrogen bond to TRIM21 I18 carbonyl (Fig. 2c). The TRIM21 linchpin-residue (R55) contacts Ube2N (K94 carbonyl) and ubiquitin (Q40 sidechain and R72 carbonyl) simultaneously (Fig. 2b). However, we were particularly struck by the prominent involvement of three anionic residues: E12, E13, and D21 (Fig. 2e; Supplementary Fig. 2c). Residues E12 and D21 hold Ube2N in place via a cluster of positively charged residues (R6, R7, K10, and R14) arranged on the same RING-facing surface of Ube2N helix 1, supported by interactions with backbone carbonyls (I18 and L20) (Fig. 2c, e). The final charge residue in the motif, E13, mediates a complex hydrogen bond network with ubiquitin via K11 and the distal RING protomer via N71, which in turn hydrogen bonds with the carbonyl of

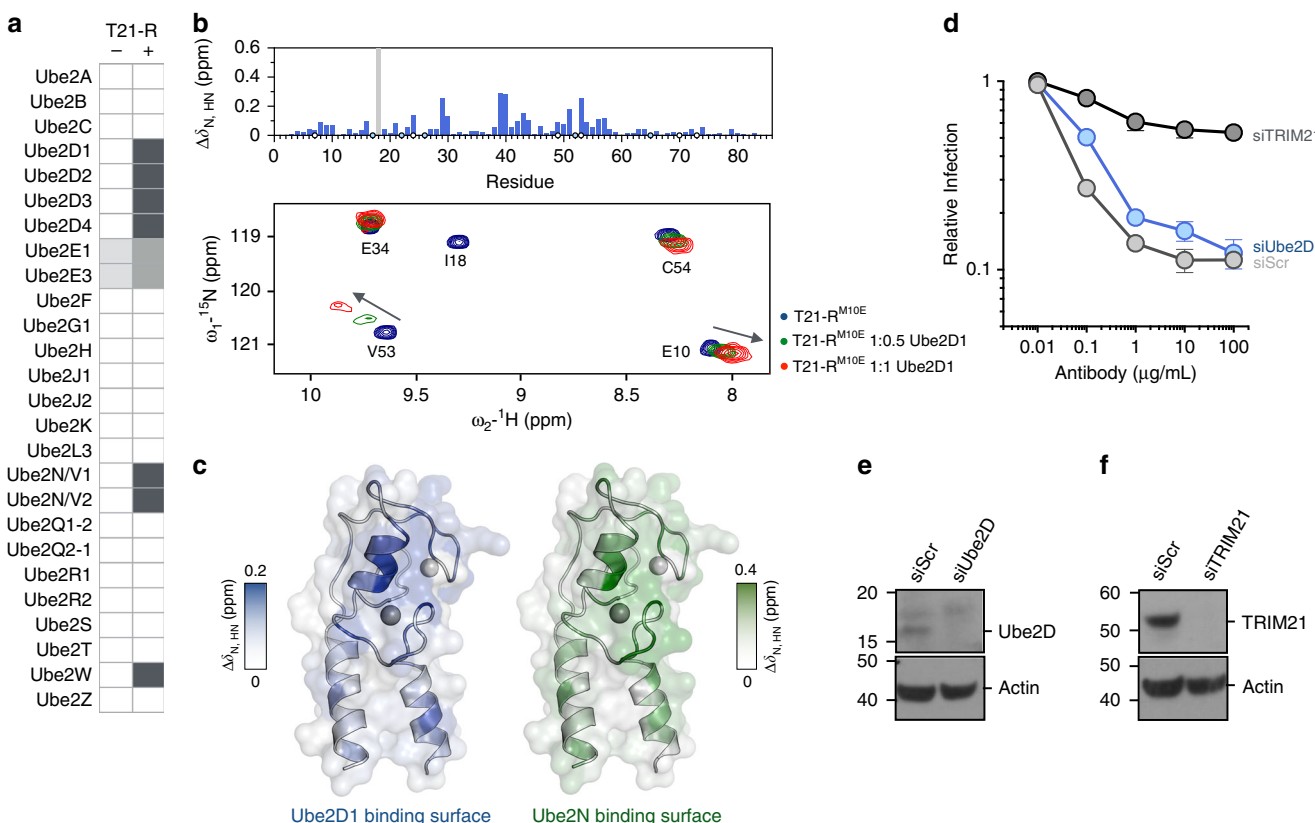

**Fig. 1** TRIM21 binds and catalyzes ubiquitination with physiologically redundant E2s. **a** In vitro E2 enzyme screen (no catalysis, white; catalysis, grey). Full blots are given in Supplementary Fig. 1. **b** A part of $^{15}$N-HSQC spectra of T21-R$^{M10E}$ in absence (blue) and presence of 0.5 (green) or 1 (red) molar equivalents of Ube2D1. Histogram of the chemical shift perturbations is shown against the primary structure. Blue circles indicate proline residues, white circles missing assignments. The gray bar in the histogram represents intermediate exchange of the amide of I18, as inferred from the disappearance of the I18 signal upon addition of Ube2D1. **c** The amide chemical shift perturbation upon E2 titration is mapped onto the T21-R structure in blue for the Ube2D1 titration (shown in Fig. 1b) and green for the Ube2N titration (shown in Supplementary Fig. 3). **d** Antibody (9C12)-dependent adenovirus 5 (Adv5) neutralization in HeLa cells stably transduced with small interfering RNA (siRNA) targeted against Ube2D or TRIM21. The data represent the mean ± s.e.m. from three independent experiments and normalized to virus only. Immunoblots of siRNA-mediated protein depletion of **e** Ube2D and **f** TRIM21. SiScr, scramble control siRNA. Source data are provided as a Source Data file

ubiquitin K33 (Fig. 2d). The tri-ionic motif is perfectly positioned to capture the E2~Ub specifically in its active, closed conformation, providing ideally distributed anchor points. Thus, in contrast to the generic interactions that are always found in E2:E3 complexes, here specific electrostatics rather than hydrophobic contacts appear to drive binding and, more crucially, specifically to the closed conformation. Additional charged associations are also used to support ubiquitin interactions. H33 of the proximal RING forms a hydrogen bond with the ubiquitin carbonyl E34, while R67 of the distal RING is positioned toward the C-terminal end of helix 1 of ubiquitin, interacting with its negative dipole moment. These latter interactions can also be found in other RING E3s; for instance the RING E3 TRAF6 bound by Ube2N~Ub[16], while TRIM25 uses a lysine residue to hydrogen bond with D32 of ubiquitin[17,18].

**Formation of the closed Ube2N~Ub conformation in solution**. To test our hypothesis that the tri-ionic motif provides an anchor point mechanism for conformation-specific capture of Ube2N~Ub, we performed NMR titrations against monomeric T21-R$^{M10E}$. The obtained amide chemical shift perturbations (CSPs) upon titration of Ube2N into $^{15}$N-labeled T21-R$^{M10E}$ (Supplementary Fig. 3; Fig. 1c) agree well with the crystal structure described above. In the crystal structure, the TRIM21

di-glutamates form salt bridges with both Ube2N and ubiquitin (Fig. 2e), suggesting distinctive roles for the two residues in this process. The E13A mutation causes no significant reduction of CSP during titration of Ube2N into $^{15}$N-labeled T21-R$^{M10E/E13A}$, whereas mutation of E12 to alanine or arginine does (Supplementary Fig. 3a, b). To quantify these interactions, we used the CSPs to calculate a $K_D$ for T21-R$^{M10E}$ binding to Ube2N of 15 ± 4 μM (mean ± s.d. of five different peaks that were fitted; Supplementary Fig. 3c), similar to the $K_D$ between the RING E3 TRAF6 and Ube2N[19]. Mutation E12R led to a fourfold increase in $K_D$ to 57 ± 13 μM, indicating impaired but not abolished Ube2N recruitment.

Based on the crystal structure, Ube2N~Ub should bind T21-R with higher affinity than Ube2N, due to the additional ubiquitin interactions (Fig. 2). Such behavior was observed for BIRC7 and Ube2D2~Ub[9]. To investigate this, we carried out a series of binding experiments using surface plasmon resonance (SPR), which allowed us to measure binding to the wild-type T21-R. We first measured a $K_D$ of 43 ± 5 μM (mean ± s.d. of two independent experiments) between immobilized GST-T21-R and Ube2N (Fig. 3a), which is slightly higher than the one measured by NMR under a different experimental setup and using T21-R$^{M10E}$ (see Methods section). Repeating the SPR measurement using Ube2N~Ub, we observed an increase in affinity by nearly an order of magnitude to a $K_D$ of 5 ± 1 μM (mean ± s.d. of three

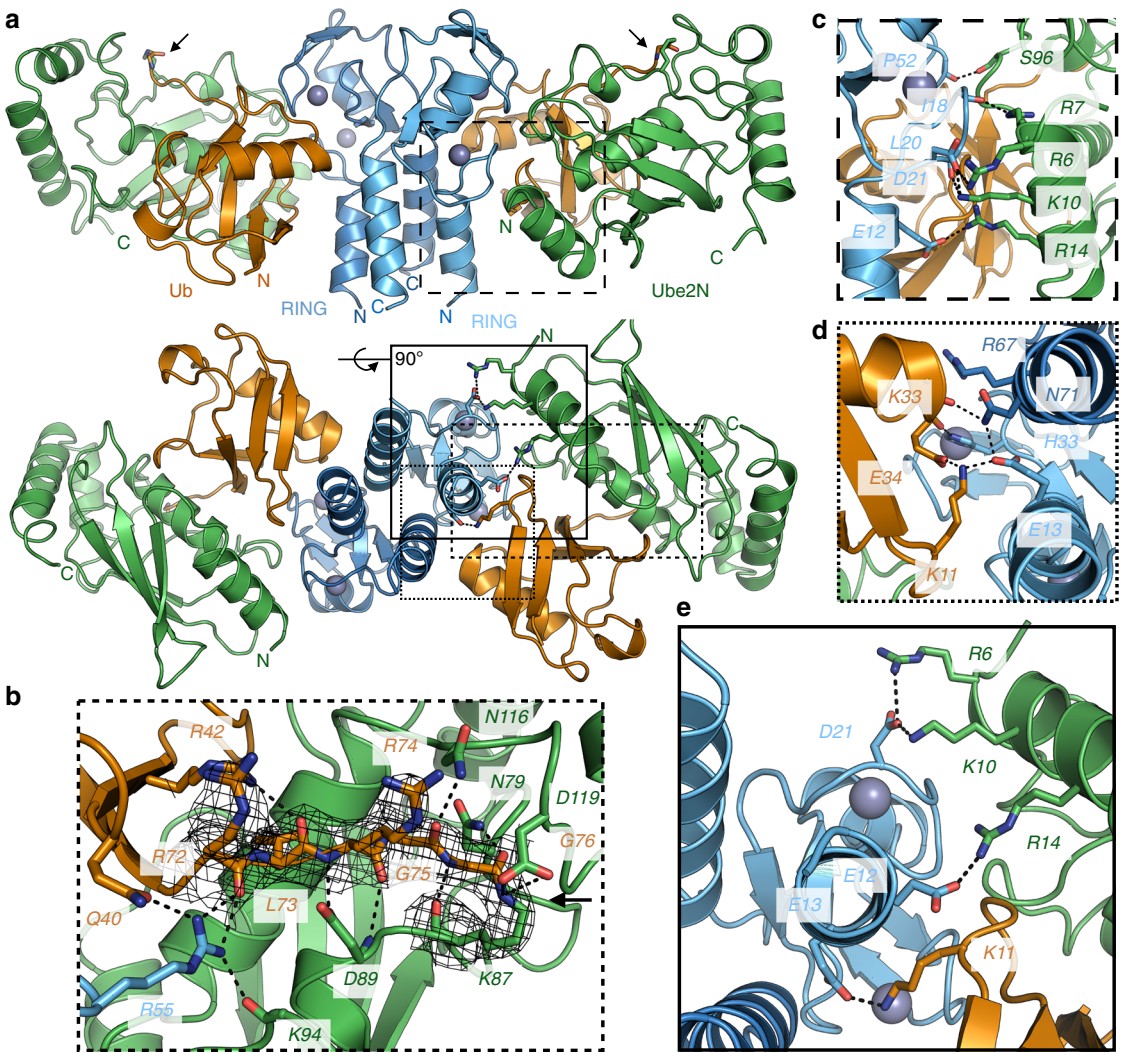

**Fig. 2** X-ray structure of TRIM21-RING in complex with Ube2N~Ub. **a** 2.8 Å X-ray structure of T21-R (blue) in complex with Ube2N~Ub (Ube2N in green, Ub in orange). Close-ups of **b** the ubiquitin C-terminus (with the $2F_O–F_C$ density at 2.0 sigma), **c** the RING:E2 interface, **d** the RING:Ub interface, and **e** the ionic anchor points. $Zn^{2+}$-atoms are shown as gray spheres, the isopeptide is marked by an arrow, and polar interactions are indicated by dashed black lines

independent experiments; Fig. 3b). RING E3s enable catalysis by promoting the closed E2~Ub conformation[8–10]. To observe formation of the closed conformation in solution, we titrated $^{15}N$-T21-R$^{M10E}$ with Ube2N~Ub. Comparison with the titration of uncharged Ube2N revealed a small number of amide peaks with a different magnitude and direction of CSP (Fig. 3c, d). We calculated the differential chemical shift perturbation (dCSP, see Methods section) between the peaks of TRIM21 in the presence of either Ube2N or Ube2N~Ub in order to isolate the effects due to ubiquitin. The amides showing the strongest dCSP values are E12, V53, C54, and R55 (Fig. 3c), in agreement with the two interfaces between ubiquitin and the proximal RING observed in the crystal structure (Fig. 2). The distal interface cannot be observed in these experiments, as monomeric T21-R$^{M10E}$ was used. To confirm formation of the closed conformation in these experiments, we also titrated T21-R$^{M10E}$ into Ube2N~$^{15}N$-Ub (Supplementary Fig. 5). Indeed, amide CSPs could be observed on ubiquitin at interfaces present in the closed Ube2N~Ub conformation (Supplementary Fig. 5d–f).

Mutation of E13, which directly stabilizes ubiquitin in the closed conformation (Fig. 2d, e), to alanine led to a strong reduction of dCSP on TRIM21 (Fig. 3e). The same was observed

when T21-R$^{M10E/E12A}$ was used, and no dCSP could be observed in case of T21-R$^{M10E/E12R}$. This latter result was unexpected as E12 contacts Ube2N, not ubiquitin, in our crystal structure (Fig. 2c, e). This suggests that the interaction between E12 and Ube2N R14 might be important for positioning E13 toward ubiquitin K11, thereby explaining the effect of E12 mutation on formation of the closed conformation. In addition, these results suggest that the anchor point residues enable catalysis by monomeric T21-R, as they enable a single RING to bind the closed conformation.

**A tri-ionic motif determines Ube2N-specificity.** Together, the X-ray and NMR data suggest that a tri-ionic motif drives TRIM21 catalysis, by specifically stabilizing an Ube2N~Ub closed conformation. To test this, we mutated the motif residues in T21-R and determined their impact on Ube2N~Ub discharge and formation of free K63-linked ubiquitin chains with Ube2N/V2. All mutants of the tri-ionic motif show a significant reduction in both activities, as does mutation of the R55 linchpin (Fig. 4a–c). Mutation of E12 to alanine greatly reduced, and mutation to arginine completely abolished, catalysis with Ube2N. Similarly, T21-R$^{E13A}$ showed a reduced activity, comparable with that

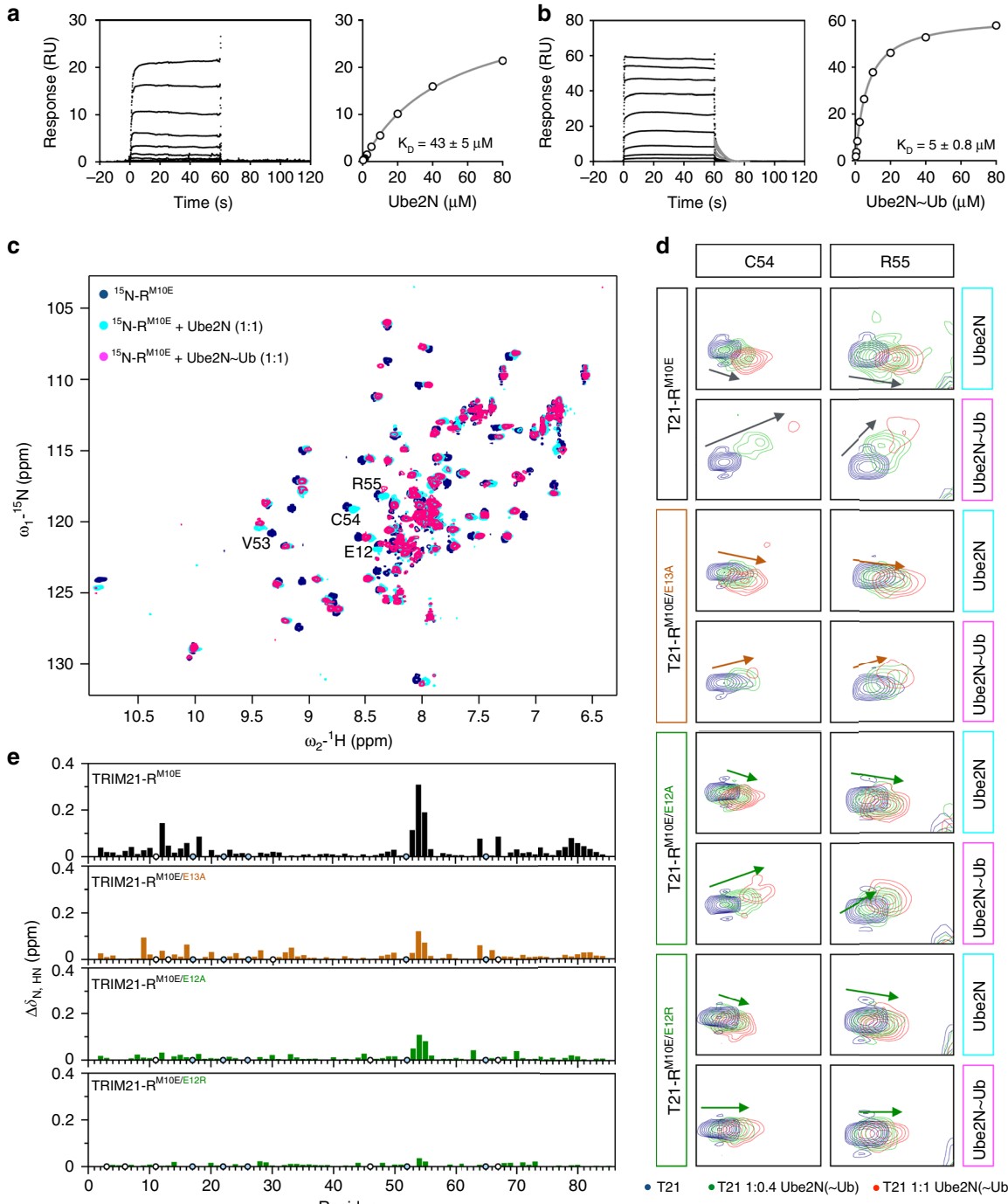

**Fig. 3** The tandem glutamates cooperatively promote formation of the closed Ube2N~Ub conformation. SPR sensograms and binding plots for immobilized GST-T21-R titrated against **a** Ube2N and **b** Ube2N~Ub. The dissociation of Ube2N~Ub could be fitted (gray) with dissociation constant ($k_{off}$) of 0.15 s$^{-1}$. For Ube2N, the dissociation was faster than the instrument response with the response returning to baseline at the end of injection, suggesting $k_{off}$ to be >1 s$^{-1}$. The data of replicates are shown in Supplementary Fig. 4. $K_D$ values represent the mean ± s.d. of two (Ube2N) or three (Ube2N~Ub) independent experiments. **c** $^{15}$N-HSQC spectra of labeled T21-R$^{M10E}$ in alone (blue), in presence of Ube2N (cyan), or Ube2N~Ub (magenta) are shown. **d** The peaks of amide C54 and R55 are shown during titration with Ube2N and Ube2N~Ub. Free T21-R$^{M10E}$ (blue), T21-R$^{M10E}$ in presence of 0.4 (green) and 1 (red) molar equivalents of titrant. **e** Histograms of the dCSP (differential chemical shift perturbation; see Methods section) of T21-R$^{M10E}$ mutants titrated with either Ube2N or Ube2N~Ub are shown against the primary structure. Blue circles indicate proline residues, white circles missing assignments. Full spectra are shown in Supplementary Fig. 8. Source data are provided as a Source Data file

observed for T21-R$^{E12A}$. T21-R$^{E13R}$ activity was reduced further. Consistent with the role of E13 in capturing ubiquitin by binding K11, the ability of T21-R to catalyze the formation of free ubiquitin chains was abolished when Ub$^{K11E}$ was used (Fig. 4d). Mutation of D21 to arginine also resulted in a loss of activity

(Fig. 4b, c). These ubiquitination assays correlate well with the NMR titrations, where E12A and E13A each show a similar reduction in dCSP and E12R shows no dCSP at all (Fig. 3e). Moreover, these biochemical results establish the tri-ionic anchor points as key residues in catalyzing ubiquitination with Ube2N.

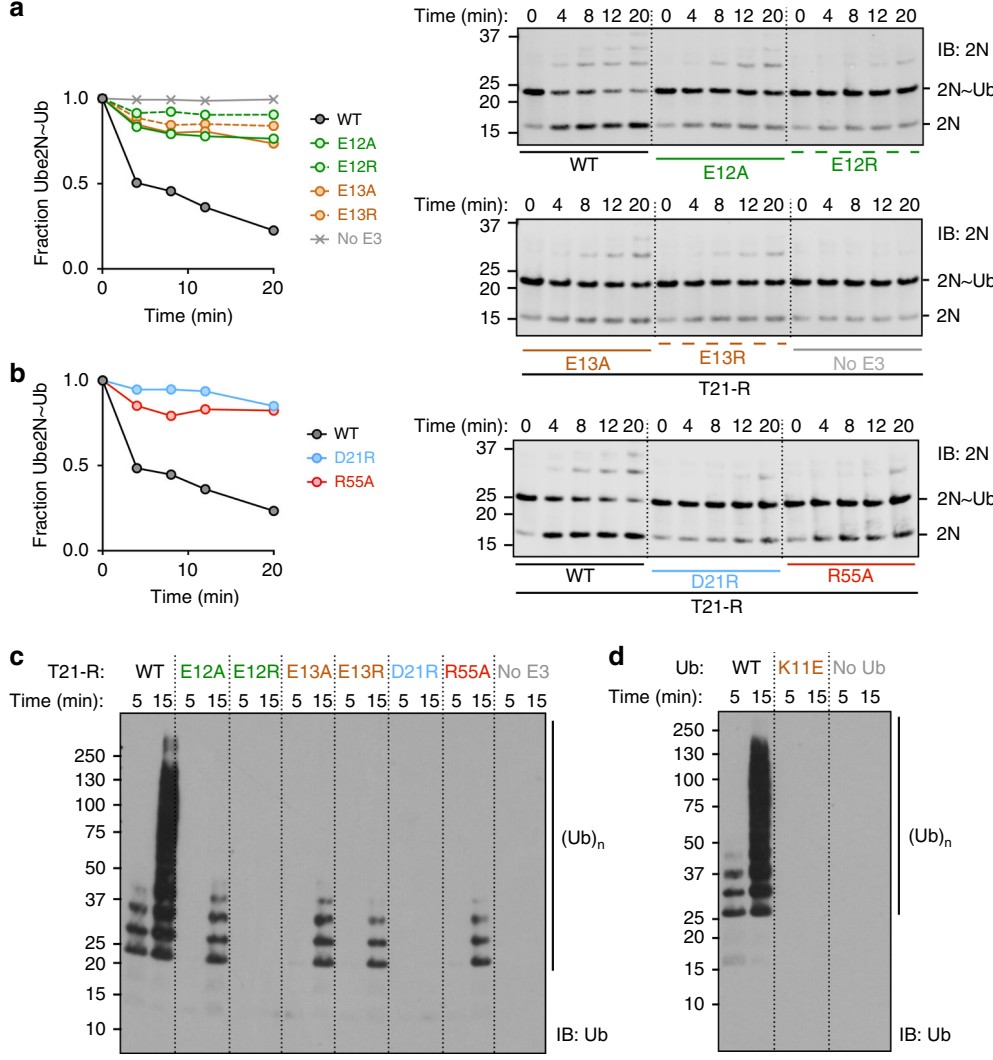

**Fig. 4** The tri-ionic anchor points are essential for K63-ubiquitination in vitro. Catalysis of ubiquitin discharge from ubiquitin-conjugated Ube2N/Ube2V2 by **a** TRIM21 di-glutamate and **b** D21R, R55A mutants. Catalysis of unanchored ubiquitin chains by Ube2N/Ube2V2 of **c** T21-R mutants and **d** ubiquitin K11E. All biochemical assays were performed in at least two independent experiments. Source data are provided as a Source Data file

A sequence alignment of the E2 enzymes identified in our screen to interact with TRIM21 shows that the charged residues in Ube2N that are coordinated by E12 and D21 are not conserved (Fig. 5a). Ube2N residue R14, which interacts with E12 is an acidic residue in Ube2D1 to Ube2D3, Ube2E1, and Ube2E3. This suggests that the tri-ionic motif has specifically evolved to promote Ube2N catalysis. To test this hypothesis, we investigated how mutants of the tri-ionic motif impact Ube2D1 catalysis by TRIM21. Strikingly, none of the tri-ionic mutants (E12A/R, E13A/R, and D21R) had any significant effect on activity with Ube2D1 (Fig. 5b, c). In contrast, linchpin mutation (R55A) killed catalysis. These results suggest that while the linchpin residue is required for efficient E2~Ub binding (Fig. 2b), it does not determine specificity. This is consistent with its role in discharging ubiquitin, which is coupled to all E2s using a common mechanism. In addition, we also swapped the charges on the E2 enzymes to make Ube2N$^{R14D}$ and Ube2D1$^{D12R}$. Consistent with the ubiquitination data on tri-ionic mutants (Fig. 4a–c) and the importance of this motif in Ube2N activity, Ube2N$^{R14D}$ displayed a strong loss of ubiquitination with T21-R (Fig. 5d). We also tested whether T21-R$^{E12R}$ could rescue this phenotype, which was not the case. Although swapping the charges could theoretically re-form the salt bridge, the

neighboring residues are different, and thus the pK$_a$ of the swapped residues will not be re-generated; indeed the pK$_a$ of a residue can be altered by charges up to 15 Å distant[20]. In contrast, Ube2D1$^{D12R}$ remained active, consistent with it not requiring the salt bridge for interaction (Fig. 5e). Collectively, these data establish that the tri-ionic motif drives a Ube2N-specific catalytic mechanism, that is distinct from catalysis with Ube2D. Significantly, Ube2W, the only other E2 known to be physiologically relevant for TRIM21 function[2], has an alanine at the structurally equivalent R14 position, which would not repel binding like the aspartate in Ube2D1.

**Tri-ionic mutants have impaired cellular function.** The identification of an E2-specific RING mechanism provided us with an opportunity to test whether TRIM21 dependence on Ube2N for antiviral activity is due to direct recruitment of the E2 or because an additional ligase is involved. To this end, TRIM21 was ectopically expressed in TRIM21-knockout 293T cells under its natural promoter and tested for antibody-dependent virus neutralization, NF-κB-mediated immune signaling and IKK protein depletion by Trim-Away (Fig. 6). In agreement with our biochemical data, all mutants of the tri-ionic motif led to reduced

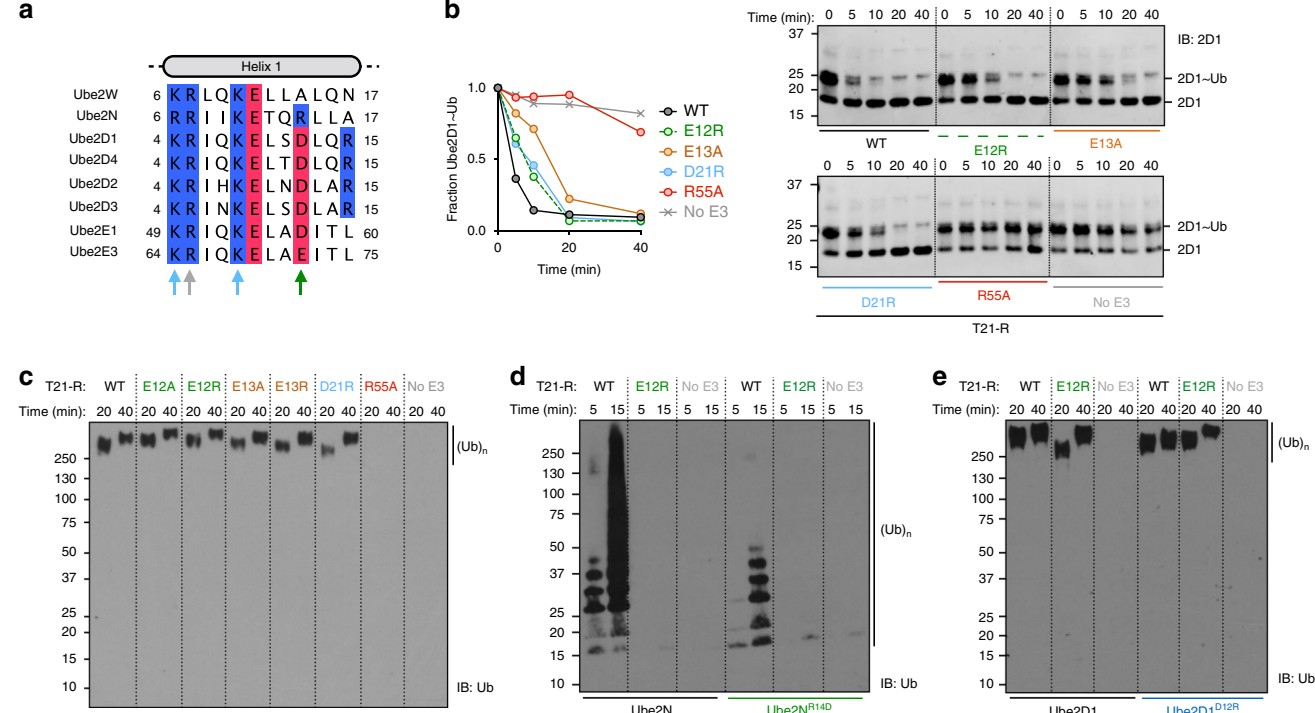

**Fig. 5** Ionic anchor points enable TRIM21 with a Ube2N-specific catalytic mechanism. **a** Clustal omega sequence alignment of all positive hits of the E2 screen (Fig. 1a). Arrows mark residues involved in RING interactions in our T21-R Ube2N~Ub structure. Lysine and arginine residues are colored in blue, aspartate and glutamate residues in red. The full alignment of all E2 enzymes used in the E2 screen (Fig. 1a) is shown in Supplementary Fig. 9. **b** Catalysis of ubiquitin discharge from pre-charged Ube2D1~Ub, and **c** catalysis of unanchored ubiquitin chains by Ube2D1 of T21-R anchor point mutants. Catalysis of ubiquitin chains by **d** Ube2N WT and R14D mutant and **e** Ube2D1 WT and D12R mutant. All biochemical assays were performed in at least two independent experiments. Source data are provided as a Source Data file

virus neutralization to varying degrees (Fig. 6a). Moreover, they all were deficient in immune signaling apart from E12A (Fig. 6b), which was the most active of these mutants in virus neutralization (Fig. 6a). All other tri-ionic motif mutants showed no immune activation. *TRIM21*[R55A] showed efficient virus neutralization and immune signaling (Fig. 6a, b). In addition, we tested the importance of these mutants in allowing TRIM21 to mediate protein depletion during Trim-Away[7]. Only *TRIM21*[E12A] was capable of efficiently depleting protein, with *TRIM21*[D21R] retaining partial depletion activity (Fig. 6c). All other mutants, including *TRIM21*[R55A], were deficient in protein depletion. Collectively, the biochemical and cellular data establishe the tri-ionic motif as a critical determinant of ubiquitination-dependent cellular function of TRIM21.

**Anchor points drive Ube2N-specificity in diverse RINGs.** Residues E12 and E13 belong to a di-glutamate motif that is present in many TRIM proteins[21] and has been suggested to be involved in ubiquitin transfer[17], though a mechanistic explanation for its conservation has not been given. Our analysis suggests that the di-glutamates are actually part of a larger conserved tri-ionic motif. The third acidic residue is also highly conserved in TRIM proteins, as either aspartate or glutamate (Fig. 7a). This suggests that the anchor point mechanism of closed conformation capture observed in TRIM21 may be driving catalytic activity of Ube2N in other TRIMs as well. To test this hypothesis, we carried out ubiquitination experiments with the retroviral restriction factor TRIM5, which utilizes the same canonical E2 enzymes as TRIM21 and has a similar mechanism of action[3]. As the RING domains of TRIM21 and TRIM5 are highly similar (Supplementary Fig. 11), we built a structural model of the catalytic complex between TRIM5-RING and Ube2N~Ub (Fig. 7b;

Supplementary Fig. 11b), by superposing a TRIM5-RING Ube2N (4TKP[22]) and our T21-R Ube2N~Ub structure (Fig. 2). This suggests that the tri-ionic motif in TRIM5 might drive ubiquitination activity with Ube2N. We first tested whether TRIM21 and TRIM5 recruit Ube2N in a similar manner, by performing an NMR titration of monomeric [15]N-TRIM5-RING[I76E] (T5-R[I76E]) with Ube2N and obtained a highly similar pattern of CSP to the equivalent titration of T21-R[M10E] (Supplementary Figs. 3a, 11c). Next, we carried out free ubiquitin chain formation and Ube2-N~Ub discharge assays with T5-R mutants corresponding to the mutants tested for T21-R. T5-R[E11R] (E12 in T21), which we predicted is required to form a salt bridge with Ube2N R14, had no apparent activity (Fig. 7c, d), in line with a previous report[11]. The second predicted ionic interaction between the E2 and RING was Ube2N R6 to T5 E20 (D21 in T21), this mutant (T5-R[E20R]) was also catalytically inactive. Finally, a mutant of the third motif residue, T5-R[E12A] (E13 in T21), did not show any catalysis, consistent with its predicted importance in stabilizing the closed conformation. These data suggest that the anchor point mechanism observed in TRIM21 is used in other TRIMs.

We further analyzed published high-resolution crystal structures of RING domains in complex with either Ube2N~Ub or Ube2N (Fig. 8). In TRIM21, TRIM5[22], and TRIM25[18], the tri-ionic anchor points are conserved both in sequence and structure. Consistent with the results for TRIM5 and TRIM21, TRIM25 mutant E10R (E13R) renders it inactive with Ube2N[17]. We could also identify anchor points in some RING E3s outside the TRIM family, including TRAF6, LNX1, ZNRF1, and CHIP (Fig. 8)[16,23–25]. Notably, only the RNF4 Ube2N~Ub structure lacks these interactions[26]. Of the identified RING E3s, in each case residue R6 at the N-terminal end of Ube2N helix 1, interacts with the first anchor point (D21 in T21). This residue is acidic in most of the

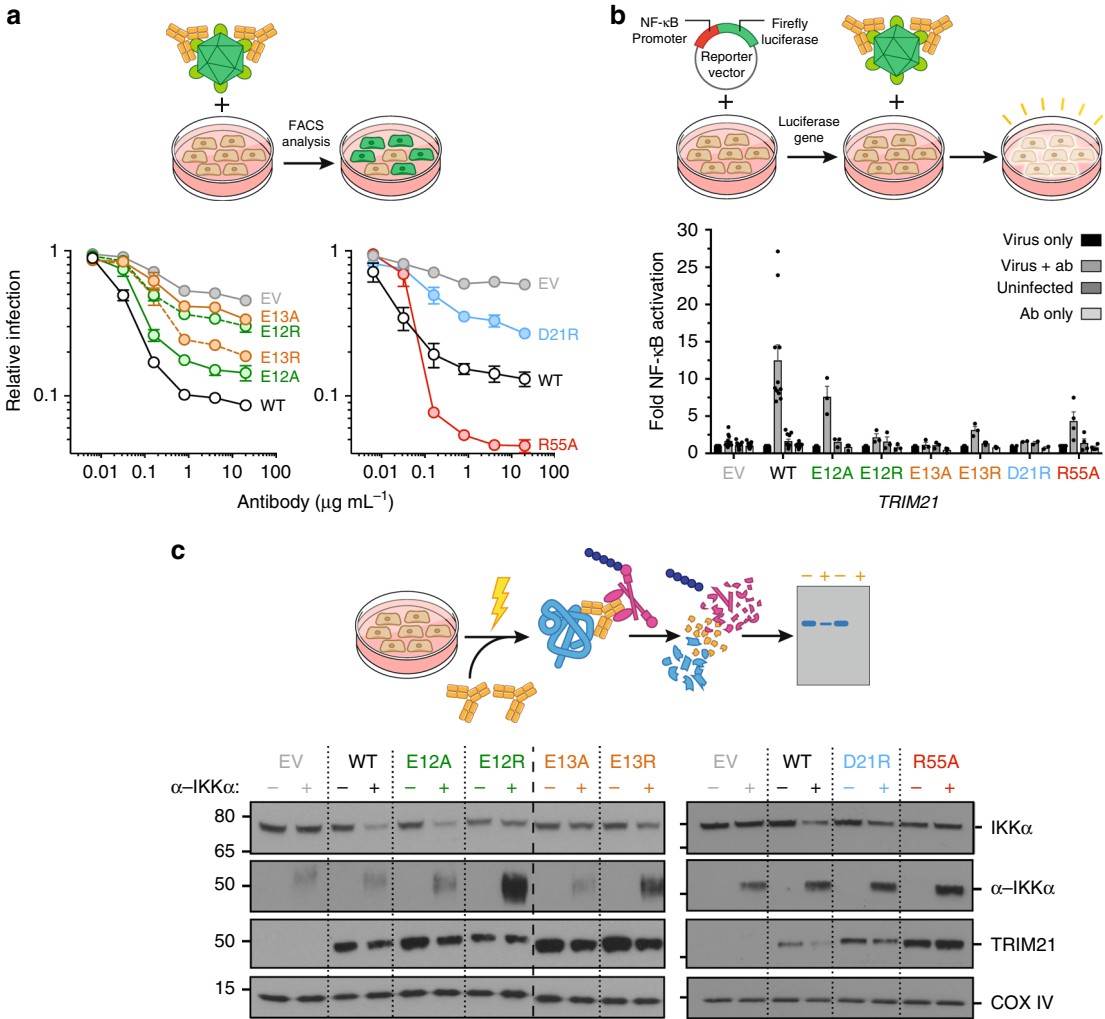

**Fig. 6** The anionic anchor point motif is essential for TRIM21 function inside cells. **a** Antibody (9C12)-dependent adenovirus 5 (Adv5) neutralization in stably reconstituted 293T cells. The data represent the mean ± s.e.m. from $n$ independent biologically experiments ($n$ (left): EV, 11; WT, 11; E12A, 4; E12R; E13A, 4; E13R, 4; D21R, 3; n(right): EV, 3; WT, 3; R55A, 3) and normalized to virus only. **b** Induction of NF-κB signaling in stably reconstituted 293T cells upon infection by 9C12-coated Adv5 measured using NF-κB luciferase reporter assay. The data represent the mean ± s.e.m. from $n$ independent biologically experiments ($n$: EV, 11; WT, 11; E12A, 3; E12R, 3; E13A, 3; E13R, 3; D21R, 2; R55A, 4) and presented as fold change over virus only. **c** Immunoblots showing Trim-Away depletion of IκB kinase alpha (IKKα) upon electroporation of 293T cells with anti-IKKα IgG (α-IKKα). Cartoon sketches explaining the cellular experiments are shown with the experimental data. The expression levels of TRIM21 can are shown in Supplementary Fig. 10a-c. EV, empty vector. Source data are provided as a Source Data file

RINGs (Q side chain and Y carbonyl in LNX1 and I carbonyl in CHIP). At the other end of Ube2N helix 1, residue R14 is always positioned to allow a salt bridge to an acidic RING sidechain (E12 in T21). The third anchor point, which contacts the ubiquitin, is found in slightly differing variations in different RINGs. LNX1 uses two aspartates (D26 and D28) to form salt bridges with K11 of ubiquitin[24], while a potential third anchor point equivalent to E13 in TRIM21 may be present in the U-box CHIP (E289, Supplementary Fig. 12). The exception is TRAF6, where ionic interactions stabilize the dimer while an adjacent Zn-finger domain interacts with the charged ubiquitin[16]. This suggests that although anchor point catalysis is a prominent feature of TRIM RINGs, a similar mechanism can be found in non-TRIM RINGs.

## Discussion

Mammalian cells typically possess ~40 E2s and ~600 E3s, giving ~24,000 theoretical E2:E3 pairs. Commonly used in vitro ubiquitination assays, together with the observed similarities between solved E2:E3 complexed structures, support the notion of significant promiscuity, yet in cells E2:E3 interactions are highly specific, delivering specialized phenotypes and functions. Determining the mechanisms behind E2:E3 specificity remains a crucial and fundamental problem in ubiquitin biology. This is because although the basis of E2 ubiquitin-chain specificity is well-understood, for instance how Ube2N builds K63-linked chains[26,27], the specificity of E3s is not. In the case of the atypical RING:E2 pair FANCL:Ube2T, a specific network of polar interactions explains their exclusivity[28], while additional modification in the form of N-terminal acetylation of the Nedd8-E2 enzyme Ube2M (Ubc12) is required for efficient interaction with the accessory E3 Dcn1[29]. However, most E2 enzymes can interact with numerous E3s, making it very difficult to identify unique features[12]. In addition, most mechanistic studies have been performed with the promiscuous Ube2D family, thereby limiting insights into specificity. Compounding the problem of E2 substrate specificity is the fact that RING E3s are also unusual in their catalysis. They do not have a classical active site, instead

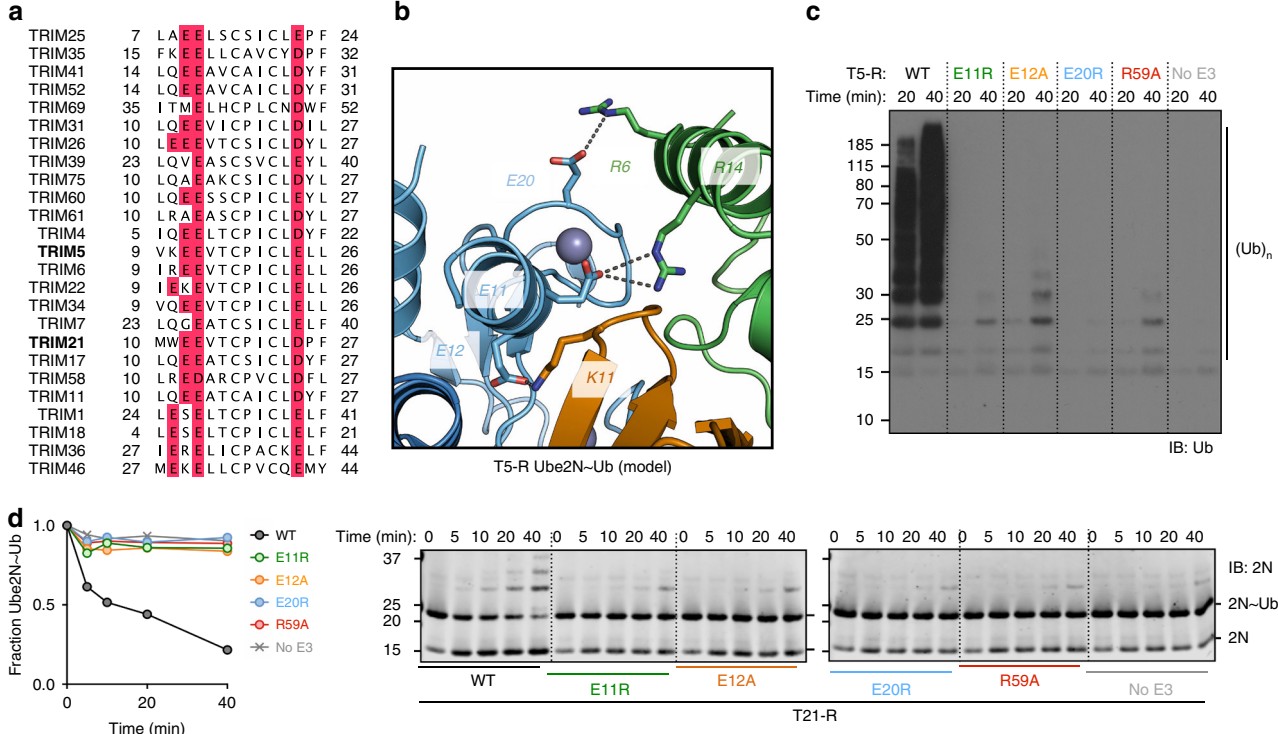

**Fig. 7** The ionic anchor points are structurally and functionally conserved in TRIM 5. **a** Clustal omega multiple sequence alignment of TRIM proteins. Ionic anchor points are highlighted in red. The sequences of TRIM21 and TRIM5 RING domains are highly similar (sequence identity: 47.3, sequence similarity: 63.4 in pairwise sequence alignment; Supplementary Fig. 6). **b** Close-up of the tri-ionic motif in the structural model of TRIM5-RING Ube2N~Ub complex based on superposition of the T5-R Ube2N (4TKP)[22] and our T21-R Ube2N-Ub structure (the full model is shown in Supplementary Fig. 11b). **c** Catalysis of unanchored ubiquitin chains by Ube2N/Ube2V2 of T5-R mutants. **d** Catalysis of ubiquitin discharge from ubiquitin conjugated Ube2N/Ube2V2 of T5-R mutants. Source data are provided as a Source Data file

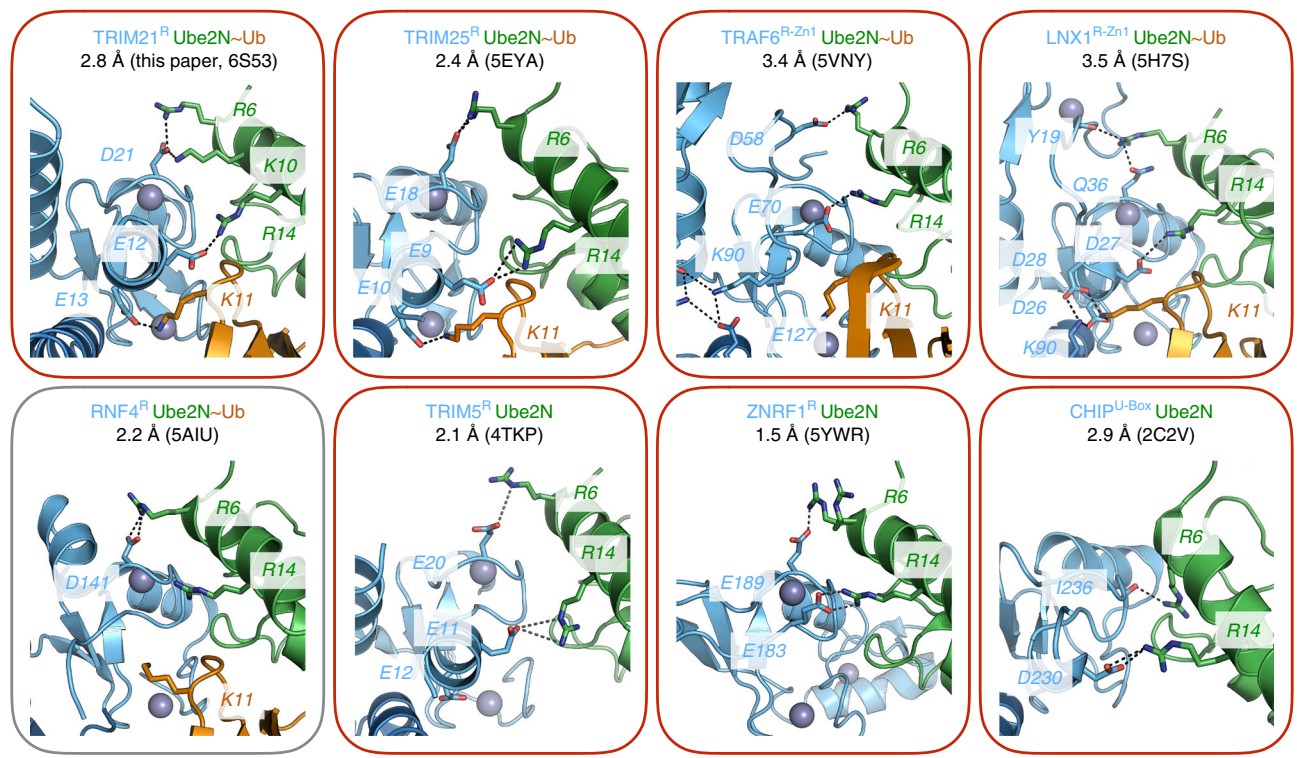

**Fig. 8** The ionic anchor points are structurally conserved outside TRIM RINGs. Close-ups of RING:Ube2N(~Ub) structures. Residues equivalent to the ionic anchor points in TRIMs are labeled. Red color of the frame indicates the presence and gray color the absence of the ionic anchor points in each structure

promoting ubiquitin discharge from E2s by capturing a closed E2~Ub conformation[8–10], akin to a transition state. Two features have been shown to be critical—a linchpin residue (R55 in T21) and RING dimerization. Crucially, the linchpin does not provide any E2 specificity because it contacts the end of the charged ubiquitin, which is coupled to almost all E2s in an identical way. In addition, a recent structure of ARC2C in complex with Ube2D2~Ub showed that the linchpin is not sufficient for formation of a catalytic complex[30]. RING dimerization is also important for catalysis, as inhibition of RING dimerization commonly abrogates RING activity for dimeric RINGs[9,17,18,22,31]. However, dimerization has not been shown to determine E2 specificity, and several active monomeric RINGs have been identified[32,33].

Here, we have described a general E2-specific mechanism of RING E3 catalysis, in which a tri-ionic motif provides conformation-specific anchor points that capture the Ube2N~Ub closed conformation. Initially described in the context of TRIM21, we show that the same mechanism is used in TRIM5 and is structurally conserved in other TRIMs and other RING E3s (Figs. 7, 8). In TRIM21, this tri-ionic motif consists of E12, E13, and D21. These residues provide spatially conserved anchor points that allow an Ube2N~Ub to be wrapped around a RING E3, thereby locking the closed conformation and promoting ubiquitin discharge. D21 forms salt bridges with R6 and K10 of Ube2N at one end of the long helix at the N-terminus of Ube2N. The first glutamate of the di-glutamate repeat forms a salt bridge with R14 at the other end of the helix. Finally, the second glutamate forms a salt bridge with ubiquitin via K11 (Fig. 2e). Importantly, in cells, mutation of anchor point residues prevents efficient TRIM21-dependent virus neutralization, immune signaling, and Trim-Away (Fig. 6). Moreover, this motif is at least as crucial as the linchpin for TRIM21 cellular function, since anchor motif mutants prevent neutralization and signaling activity, but a linchpin mutant is still active (Fig. 6).

A crucial feature of the anchor point mechanism is that it specifically drives catalysis of Ube2N, and not that of other E2 enzymes such as the promiscuous E2 Ube2D (Figs. 4, 5). While mutation of the anionic residues in both TRIM5 and TRIM21 strongly reduces ubiquitination with Ube2N in vitro, it does not affect activity of TRIM21 with Ube2D1 (Fig. 5). Classically, hydrophobic interactions and π-stacking have been described as the driving force behind E2:E3 interaction, but it is the charged nature of the interactions mediated by the anchor motif that enables it to promote activity with Ube2N and not Ube2D1. This is because while Ube2N has a positively charged residue positioned to interact with E12, the structurally equivalent residue in Ube2D is a negatively charged aspartate. This probably explains why an E9R TRIM25 mutation increased rather than decreased activity with Ube2D1[17]. TRIM21 cellular function is dependent upon the E2 enzymes Ube2W and Ube2N/Ube2V2, which act sequentially to build a TRIM21-anchored K63-linked ubiquitin chain[2,5]. By identifying a Ube2N-specific catalysis mechanism and introducing specific mutations that ablate it in cells, we have been able to show that the requirement for Ube2N in TRIM21 function is because of direct recruitment. A similar strategy could now be used in order to test whether direct Ube2N recruitment is necessary for the function of other RING E3s during their cellular activity.

Identification of the tri-ionic motif may also explain why monomeric T21-R is capable of performing catalysis. All three residues are located in one RING monomer and are critical for capture of the E2~Ub complex; specifically, mutation of E12 or E13 strongly reduced formation of the closed conformation in solution (Fig. 3). However, while the anchor points provide a mechanism that allows T21-R to be active as a monomer, they are

not sufficient to drive monomeric activity in TRIM5[22]. This may be because TRIM21 has evolved to become a significantly more potent ligase than TRIM5[13]. Increased RING activity may also explain the importance of B-box autoinhibition in controlling TRIM21 function. While we have identified a Ube2N-specific mechanism, it remains an open question why TRIM21 (or indeed other RING E3s) does not seem to utilize Ube2D in cells, despite being able to. Spatio-temporal differences in protein expression and availability may contribute to functional specificity. For instance, in HeLa and U2OS cells, the Ube2N concentration is three to six times higher than for Ube2D (Supplementary Fig. 13)[34,35]. It will be interesting to determine whether there are other general mechanisms, such as that described here, that allow specific E2 recruitment by E3 ligases.

## Methods

**Expression and purification of recombinant protein**. TRIM21-RING (residue 1–85), TRIM5-RING (1–88), Ube2N, and Ube2D1 constructs were expressed in *Escherichia coli* C41 DE3 or BL21 DE3 cells. Ubiquitin and Ube1 were expressed in *E. coli* Rosetta 2 DE3 cells. All cells were grown in 2xTY media supplemented with 2 mM MgSO$_4$, 0.5% glucose, and 100 µg mL$^{-1}$ ampicillin. Cells were induced at an OD$^{600}$ of 0.7. For TRIM proteins, induction was performed with 0.5 mM IPTG and 10 µM ZnCl$_2$, for ubiquitin and Ube1 with 0.2 mM IPTG. After centrifugation, cells were resuspended in 50 mM Tris pH 8.0, 150 mM NaCl, 10 µM ZnCl$_2$, 1 mM DTT, 20% Bugbuster (Novagen), and complete protease inhibitors (Roche, Switzerland). Lysis was performed by sonication. TRIM proteins were expressed with the N-terminal GST-tag and purified via glutathione sepharose resin (GE Healthcare) equilibrated in 50 mM Tris pH 8.0, 150 mM NaCl, and 1 mM DTT. The tag was cleaved on beads overnight at 4 °C. Ube2N, Ube2D1, and Ube1 were expressed with an N-terminal His-tag, and were purified via Ni-NTA resin. Proteins were eluted in 50 mM Tris pH 8.0, 150 mM NaCl, 1 mM DTT, and 400 mM imidazole. For Ube2N, TEV-cleavage of the His-tag was performed overnight. For Ube2D1, no cleavage was performed. The cleavage left an N-terminal tripeptide scar (GSH) on recombinantly expressed TRIM proteins and an N-terminal G scar on Ube2N. Finally, size-exclusion chromatography was carried out on a HiLoad 26/60 Superdex 75 prep grade column (GE Healthcare) in 20 mM Tris pH 8.0, 150 mM NaCl, and 1 mM DTT. Ubiquitin purification was performed following the protocol established by the Pickart lab[36]. After cell lysis by sonication (lysis buffer: 50 mM Tris pH 7.4, 1 mg mL$^{-1}$ Lysozyme (by Sigma-Aldrich, St. Louis, USA), 0.1 mg mL$^{-1}$ DNAse (by Sigma-Aldrich, St. Louis, USA)), a total concentration of 0.5% percloric adic was added to the stirring lysate at 4 °C. The (milky) lysate was incubated for another 30 min on a stirrer at 4 °C to complete precipitation. Next, the lysate was centrifuged (19,500 rpm) for 30 min at 4 °C. The supernatant was dialyzed overnight (3500 MWCO) against 3 L 50 mM sodium acetate. Afterward, Ub was purified via cation-exchange chromatography using a 20 mL SP column (GE Healthcare) using a NaCl gradient (0–1000 mM NaCl in 50 mM sodium acetate pH 4.5). Finally, size-exclusion chromatography was carried out on a HiLoad 26/60 Superdex 75 prep grade column (GE Healthcare) in 20 mM Tris pH 7.4. Isotopically labeled protein was expressed using *Escherichia coli* BL21 DE3 cells (TRIM proteins) or *E. coli* Rosetta 2 DE3 cells (ubiquitin) in M9 minimal media supplemented with either $^{15}$NH$_4$Cl or $^{15}$NH$_4$Cl and [$^{13}$C$_6$]glucose (Sigma-Aldrich ISOTEC).

**Formation of an isopeptide-linked Ube2N-Ub**. Ube2N$^{C87K/K92A}$ charging with WT ubiquitin was based on a protocol of the Hay-lab[26]. The isopeptide charging reaction occurred in 50 mM Tris pH 10.0, 150 mM NaCl, 5 mM MgCl$_2$, 0.5 mM TCEP, 3 mM ATP, 0.8 µM Ube1, 100 µM Ube2N, and 130 µM ubiquitin at 37 °C for 4 h. After conjugation, Ube2N$^{C87K/K92A}$~Ub was purified by size-exclusion chromatography (Superdex S75 26/60, GE Healthcare) that was equilibrated in 20 mM Tris pH 8.0 and 150 mM NaCl.

**Single turnover ubiquitin discharge assay**. Ube2N$^{K92R}$ or Ube2D1 were charged with ubiquitin by incubating 40 µM E2, 1 µM Ube1, 0.37 µM ubiquitin, and 3 mM ATP in 50 mM HEPES pH 7.5, 150 mM NaCl, 20 mM MgCl$_2$ at 37 °C for 45 min. Afterward, this charging mix was cooled at 4 °C and used within 1 h. To observe E2~Ub discharge, 2 µM E2~Ub was added to 50 mM HEPES pH 7.5, 150 mM NaCl, 20 mM MgCl$_2$, 50 mM L-lysine, and 2.5 µM Ube2V2. For TRIM21 assays with Ube2N$^{K92R}$ or Ube2D1, respectively, 1.5 µM and 1 µM T21-R were used, respectively. For TRIM5 assays, 10 µM T5-R was used. The reaction took place at 37 °C, and was initiated by addition of the E3. Samples were taken at the time points indicated, and the reaction was stopped by addition of LDS sample buffer at 4 °C. Samples were boiled for 20 s at 90 °C, resolved by LDS-PAGE and observed using immuno blot. Anti-Ube2N (BioRad, AHP974 1:5000) or anti-Ube2D (Boston Biochem, A-615, 1:2500) antibodies (and the LiCor system) were used in immuno blot analysis.

**Ubiquitin chain-formation assay.** Ubiquitination reactions were performed with 0.1 mM ubiquitin, 2 mM ATP, 1 μM Ube1, and 0.5 μM Ube2N/V2 or 0.25 μM Ube2D1 in 50 mM Tris pH 8, 2.5 mM MgCl₂, 0.5 mM DTT. The reaction took place at 37 °C, and was initiated by addition of E3. For TRIM21, 1.5 μM T21-R, and for TRIM5 10 μM T5-R constructs were used. Samples were taken at the time points indicated, and the reaction was stopped by addition of LDS sample buffer at 4 °C. The samples were boiled at 90 °C for 2 min and resolved by LDS-PAGE. Ubiquitin chains in the immuno blot were detected using an anti-Ub-HRP (Santa Cruz, sc8017-HRP P4D1, 1:10,000).

**Protein crystallization, structure solution, and refinement.** In total, 10 mg mL$^{-1}$ of T21-R with Ube2N$^{C87K/K92A}$~Ub in 50 mM deuterated Tris pH 7.0, 150 mM NaCl, and 1 mM deuterated DTT were subjected to sparse matrix screening in sitting drops at 17 °C, and crystals were initially obtained in the MORPHEUS I screen[37]. After crystal growth refinement, crystals grew in 0.1 M Tris/BICINE pH 8.5, 10.5% (w/v) PEG3350/PEG 1 K/MPD, and 0.08 M sodium nitrate/sodium phosphate/ammonium sulfate. Crystals were flash frozen in the crystallization condition supplemented with 15% glycerol.

The data were collected at the Diamond Light Source beamline i04, equipped with a PILATUS 6 M Prosport + detector at a wavelength of 0.97952 Å (Beamline i04). Diffraction images were processed using XDS[38] to 2.8 Å resolution (CC$_{1/2}$: 0.998 (0.735 for 2.8–2.9 Å)). The crystals belong to the spacegroup P1 with two complexes per asymmetric unit (2xT21-R, 2x Ube2N$^{C87K/K92A}$~Ub; Supplementary Fig. 2e). The structure was solved by molecular replacement using PHASER-MR implemented in the Phenix suite[39]. Search models served the RING domain of TRIM21$^{RING-Box}$ structure (5OLM)[13] and the Ube2N~Ub (5EYA)[18] from a complex structure with TRIM25$^{RING}$. Model building and real-space refinement was carried out in COOT[40], and refinement was performed by using phenix-refine toll in PHENIX and REFMAC5 iteratively[39,41]. The isopeptide bond between Ube2N K87 and ubiquitin G76 was created by using aceDRG[42]. Overall, 96.8% backbone dihedrals were in favored Ramachandran regions with 0.51% Ramachandran outliers.

**NMR spectroscopy.** Two-dimensional NMR measurements ($^{15}$N-HSQC and $^{15}$N-BEST-TROSY) were performed at 25 °C on Bruker Avance I and III 600 MHz spectrometers equipped with 5 mm $^{1}$H–$^{13}$C–$^{15}$N cryogenic probes. The data were processed with the program Topspin (Bruker BioSpin GmbH, Germany) and analyzed with CCPN analysis[43]. Samples were buffer exchanged into 50 mM deuterated Tris pH 7.0, 150 mM NaCl, and 1 mM deuterated DTT (Cambridge Isotopes, UK).

Chemical shift perturbations were calculated using the following equation:

$$\Delta\delta_{N,HN} = \sqrt{\left(\Delta\delta(^{1}H)\right)^2 + \left(\Delta\delta(^{15}N)^2 \times 0.14\right)} \tag{1}$$

where $\Delta\delta_{N,HN}$ is the CSP, $\Delta\delta(^{1}H)$ and $\Delta\delta(^{15}N)$ are the differences between the position of proton or nitrogen signal in absence and presence of titrant.

Differential chemical shift perturbations (dCSPs) were calculated using the same formula as for CSPs, but applied to differences between chemical shifts measured for the corresponding signal at the corresponding titration point during titrations with two different titrants.

Dissociation constants ($K_D$) of the T21-R constructs with Ube2N were determined by fitting the titration points to the following formula in Graphpad Prism 7 (GraphPad Software Inc.):

$$\Delta\delta_{obs} = \frac{\Delta\delta_{max}\left\{\left([T21]_t + [2N]_t + K_D\right) - \sqrt{\left(\left([T21]_t + [2N]_t + K_D\right)^2 - 4[T21]_t[2N]_t\right)}\right\}}{2[T21]_t}, \tag{2}$$

where $\Delta\delta_{obs}$ is the observed CSP, $\Delta\delta_{max}$ is the maximum CSP, which is obtained during fitting, $[T21]_t$ is the total concentration of T21-R, $[2N]_t$ is the total concentration of Ube2N, and $K_D$ the dissociation constant.

$^{15}$N-T5-R$^{I76E}$ and Ube2N~$^{15}$N-Ub were assigned based on standard triple-resonance spectra (HNCACB, HN(CO)CACB, HNCA, HN(CO)CA), that were recorded at 25 °C on a Bruker Avance III 600 MHz spectrometer. In the case of TRIM21, the previously published T21-R$^{M10E}$ assignments were used[13].

TRIM21 titrations with Ube2N$^{C87K/K92A}$ were performed with 100 μM T21-R$^{M10E}$. The Ube2D1 titration contained 200 μM T21-R$^{M10E}$. The TRIM5 titration with Ube2N$^{WT}$ contained 200 μM T5-R$^{I76E}$.

**Surface plasmon resonance.** Surface Plasmon Resonance (SPR) data were collected using a BIAcore T200 instrument (GE Healthcare Life Sciences) at a flow rate of 30 μl min$^{-1}$ in 20 mM Tris-HCl pH 8.0, 150 mM NaCl, 1 mM DTT at 25 °C. GST-tagged T21-R, or recombinant GST on the reference channel, were captured on an anti-GST antibody-coated CM5 sensor chip (GE Healthcare) prepared according to the supplier's instructions. Ube2N~Ub or Ube2N in 1:2 dilution series with initial concentrations of 80 μM were injected for 60 s, and dissociation monitored for 300 s. The sensor surface was re-generated after each injection with a 120 s injection of 10 mM glycine, pH 2.1. The data were doubly referenced by subtraction of the reference channel data and from injections of buffer alone. The data were fit using KaleidaGraph (Syngery Software) and Prism (GraphPad Software Inc). The rate constants of dissociation were measured by fitting dissociation

data at time $t$ ($R_{dissoc}$) using a single-exponential function:

$$R_{dissoc} = R_0 \exp^{-(k_{off}t)} + RI + Dt, \tag{3}$$

where $k_{off}$ is the dissociation rate constant, $R_0$ is maximum change in response each phase, RI is the bulk resonance change, and D is a linear drift term. The responses at equilibrium ($R_{eq}$) were fitted using a single-site binding model:

$$R_{eq} = \left(\frac{CR_{max}}{C + K_D}\right) + RI, \tag{4}$$

where $K_D$ is the dissociation constant, C is the analyte concentration, and $R_{max}$ is the maximum change in resonance.

**E2-conjugating enzyme screen.** The E2Select Ubiquitin Conjugating Kit (K-982, Boston Biochem, Cambridge, USA) was used as described in the product manual. The E3 concentration was 1.8 μM T21-R, and for detection anti-Ub-HRP (Santa Cruz, sc8017-HRP P4D1, 1:10,000) or anti-T21-R sera (1:1000[13]) was used.

**Cells.** Human embryonic kidney 293T (293T; ATCC number CRL-3216) and HeLa cells (ATCC number: CCL-2) were maintained in the Dulbecco's Modified Eagle Medium supplemented with 10% (v/v) fetal bovine serum (Gibco), 100 U ml$^{-1}$ penicillin and 100 μg ml$^{-1}$ streptomycin. Cells were cultured at 37 °C with 5% CO₂ and passaged every 2–3 days.

**Transient siRNA knockdown.** For transfection, 30 pmol of pooled small interfering (si) RNA oligonucleotides for Ube2D1, Ube2D2 and Ube2D3 (CCAA AGAUUGCUUUCACAAUU (Ube2D1), GGUGGAGUCUUCUUUCUCAUU (Ube2D2), CAGUAAUGGCAGCAUUUGU (Ube2D2 and Ube2D3, GAUCACA GUGGUCGCCUGC (Ube2D1)) were mixed with 500 μL Ptimem and 5 μL RNAi Max (Invitrogen) in one well of a six-well plate. Samples were incubated at room temperature for 20 min, then 10$^5$ cells were added to each well in 2 mL complete media. Neutralization assays were carried out 48–72 h post transfection.

**Lentiviral vector production.** Pseudotyped lentiviral vectors were produced by co-transfection of 5 × 10$^6$ WT 293T cells in 10 cm$^2$ dishes with 1 μg pCRV-GagPol, 1 μg pMD2G-VSVg, and 2 μg pNatP-TRIM21 using Fugene-6 (Promega). Viral supernatants were harvested at 48 h post transfection and filtered using 0.45 μM syringe filters.

**Generation of stable cell lines.** TRIM21 K.O 293T cells (generated for a previous study[13]) were infected with lentivirus containing supernatant at an MOI ~1 in the presence of 5 μg mL$^{-1}$ polybrene, and stably transduced cells were selected using puromycin at 2.5 μg ml$^{-1}$.

**Adenovirus neutralization assay.** For Adv5-GFP infections in siRNA knockdown experiments, HeLa cells were seeded at 1 × 10$^5$ cells per well in 2 mL complete DMEM in six-well plates the day before infection. In total, 3 × 10$^4$ infectious units (IU) of AdV5-GFP were incubated with mouse 9C12 antibody in a 10 μL volume for 30 min at room temperature before addition to cells. Cells were incubated for 48 h before washing, trypsiniation, and fixing in 4% paraformaldehyde. GFP-positive cells were enumerated by flow cytometry (FACSCalibur, BD Biosciences, San Jose, CA, USA).

In case of virus neutralization assays of TRIM21 mutants, 293T cells were plated at a density of 5 × 10$^4$ cells per well in 24-well plates and were allowed to attach overnight. For each well, 2.5 × 10$^5$ IU of Adv5-GFP (ViraQuest) were mixed 1:1 with human 9C12 antibody (at the indicated concentrations), and incubated for 1 h at 20 °C before adding to cells. In all, 10 μL of virus–antibody complex were added per well, and cells were incubated for 20 h at 37 °C. Cells were then harvested by trypsiniation and evaluated for GFP expression by flow cytometry (LSRFortessa, BD Biosciences, San Jose, CA, USA). The results were analyzed using FlowJo software (FlowJo LLC), and relative infection was calculated using the method described previously[4]. Gating was performed for GFP-positive and negative cells. Control were cells that were not infected with GFP-labeled virus (background 0.1%; Supplementary Fig. 11d).

**NF-κB signaling assay.** 293T cells were plated at a density of 5 × 10$^4$ cells per well in six-well plates a day before transfection with 250 ng of pGL4.32 NF-κB luciferase (Promega) using FuGENE 6 (Promega). Cells were incubated for 6 h at 37 °C before reseeding at a density of 1 × 10$^4$ per well in Corning® CellBIND® 96-well plates and allowed to adhere overnight. For each well, 6.25 × 10$^6$ IU of Adv5-GFP (ViraQuest) were mixed 1:1 with 20 μg mL$^{-1}$ antibody (human 9C12) and incubated for 1 h at 20 °C to allow complex formation. In total, 5 μL of the virus–antibody complex were added per well and allowed to incubate for 6 h at 37 °C before the cells were lysed with 100 μL per well steadylite plus luciferase reagent (Perkin Elmer). The luciferase activity was measured using a BMG PHERstar FS plate reader.

**Trim-Away**. Anti-IKK alpha antibody (ab169743, Abcam) was delivered into cells using the Neon Transfection system (Invitrogen). For each electroporation reaction, $8 \times 10^5$ cells suspended in 10.5 μl of Resuspension Buffer R were mixed with the indicated amount of antibody diluted in 2 μl of PBS. The cell antibody mixtures were taken up into 10 μl Neon electroporation pipette tips (Invitrogen) and electroporated using the following settings: 1400 V, 20 ms, 2 pulses (as described in ref. [7]). Electroporated cells were transferred to antibiotics-free DMEM supplemented with 10% FBS and left to incubate for 3–5 h in an incubator before the cells were harvested for immunoblotting. Cells were primed with human Interferon alpha (PeproTech) at 24 h before electroporation with a final concentration of 1000 U ml$^{-1}$.

**Antibodies**. Mouse 9C12 anti-adenovirus 5 hexon IgG was purified from hybridoma obtained from the Developmental Studies Hybridoma Bank, University of Iowa, IA, USA. Humanized anti-adenovirus hexon antibody 9C12 were produced by the Andersen lab for previous studies[44,45].

Antibodies used in immunoblots were anti-TRIM21(D-12) Santa Cruz Biotechnology (SC25351), 1:500, anti-UbcH5 Boston Biochem (A-615), 1:1000, anti-Ube2N BioRad (AHP974), 1:1000, anti-COX IV LI-COR Biosciences (926–42212), 1:5000, anti-Ub-HRP Santa Cruz (sc8017-HRP P4D1), 1:5000, anti-TRIM21 [raised against human TRIM21 RING-B-Box-Coiled Coil[4], 1:1000], anti-β-actin-HRP Santa Cruz (sc47778), 1:20,000. Secondary antibodies were anti-mouse-HRP Sigma (A0168), 1:5000, anti-rabbit-HRP Cell Signaling (7074), 1:5000. All antibodies were diluted in PBS containing 5% nonfat milk and 0.01% Tween20 or 3% BSA in 0.01% Tween20 for ECL and LI-COR visualization, respectively. Visualization was carried out using an ECL Western Blotting Detection System (GE Healthcare) or Odyssey CLx near-infrared imaging system (LI-COR Bioscience). All uncropped blots and gels are provided in the Source Data file.

**Reporting summary**. Further information on research design is available in the Nature Research Reporting Summary linked to this article.

## Data availability

The source data for Figs. 1, 3, 4, 5, 6, 7, and Supplementary Figs. 1–5, 10, and 13 are provided as a Source Data file. The crystal structure was deposited in the Protein Data Bank under the accession code 6S53. All other data are available from the corresponding author on reasonable request.

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

## Acknowledgements

We would like to thank Dr David Barford, Dr Katherine Stott, Dr Marios G. Koliopoulos, Dr Christina Gladkova, Laura E. Easton, Dr Dean Clift, Dr Minmin Yu, and Nadine Renner for valuable discussions. Dr Paul Emsley, Dr Nicolas Nugenuin-Dezot, and Dr Marios G. Koliopoulos for help with isopeptide modeling in the structure. We would also like to thank Diamond Light Source for beamtime (proposal: mx15916) and the staff of beamline i04 for assistance at the beamline. We also thank Dr Jan Terje Andersson for the kind gift of humanized 9C12 antibody and Joanne Westmoreland for the preparation of the cell biology sketches. This work was supported by the MRC (UK, U105181010 to L.C.J. and U105178934 to D.N.) and a Wellcome Trust Investigator Award to L.C.J. This work was also supported by a PhD Fellowship from the Boehringer Ingelheim Fonds to L.K., a PhD Fellowship of the Frank Edward Elmore Fund (University of Cambridge) to J.Z. and C.F.D. was supported by an NHMRC Early Career Fellowship (GNT1110116).

## Author contributions

L.K. conceptualized this study, performed and analyzed biochemical experiments, crystallized and solved the structure, acquired and analyzed NMR and SPR data, analyzed all experimental data, and wrote the paper. J.Z. conceptualized this study, performed cell biological experiments and assisted with biochemical experiments. C.F.D. contributed to conceptualization, preliminary data, and new reagents. D.L.M. performed siRNA knockdown experiments. J.-C.Y. helped acquiring and analyzing NMR data. S.H.M. acquired and analyzed SPR data. A.B. helped in crystallization and structure solution. D.N. conceptualized and supervised this study, analyzed experimental data, and contributed to the writing process. L.C.J. initiated, conceptualized, and supervised this study, analyzed all experimental data and wrote the paper.

## Competing interests

The authors declare no competing interests.
