## [Peer Review File · Nature Communications]

Reviewers' comments:

Reviewer #1 (Remarks to the Author):

This manuscript from Kiss and co-workers describes the recognition of TRIM21 for distinct E2 enzymes that result in K63 chain linkage. The authors combine a variety of ubiquitination assays and NMR experiments to identify a contact surface between Ube2N and Ube2D1 with TRIM21 (T21-R). A crystal structure of T21-R with Ube2N-Ub shows several ionic interactions between the E2 enzyme (R6, R6, K10 and R14) with T21-R (E12, E13, D21) that appear to position the E2 and Ub portions of the conjugate for catalysis. In particular, the authors show nicely that E12 of T21-R is important for Ub discharge from the conjugate for Ube2N but not for Ube2D1. Altogether the work presents some important new aspects of T21-R function and its ubiquitination activity with selected E2 enzymes. The work is generally well done and should be a useful resource to other investigators.

There are several aspects of the manuscript I feel need to be improved. Firstly, the E2 screen in Fig. 1a and Fig. S1 is rather ambiguous. The authors claim that Ube2D1-4 function with T21-R yet the blots in Fig. S1 do not show this. There is a large unresolved section of the Ub-IB that might be attributed to significant ubiquitination but this does not show up on the lower gels. In addition, one would expect that some discrete bands would be visible on the IB as they are for Ube2N/V1 and Ube2N/V2. Further, if the large blot is positive ubiquitination as the authors suggest, one would conclude that Ube2D1-4 are the most active E2s. This would not be consistent with the authors conclusions from the crystal structures. Further, if activity can not be shown one would question the use of Ube2D1 in other experiments. This assay needs to be repeated and clarified.

From the NMR data it is clear that Ube2N (Ube2D1) bind quite weakly to T21-R. From the crystal structure one would predict that binding should be enhanced in the E2~Ub conjugate. This should be quantified.

Despite the importance of E12, E13 in T21-R the CSPs in this region are minimal. This should be discussed.

The alignment in 4E should be applied to all E2s tested in Fig. 1 to show that none of the other E2 enzymes (ie. non-functional) fulfill the basic residue requirement.

On p8.9 the authors describe experiments to confirm the closed E2~Ub conformation in solution by monitoring CSPs in T21-R with Ube2N and Ube2N-Ub (Fig. 3). Although different CSPs can be noted

for these two titrations this data does not show the closed conformation but rather CSPs from E2 or E2-Ub binding. In order to monitor the closed conformation the authors would need to monitor the Ub or E2 within the conjugate as has been done by several other groups. This is a straight-forward experiment.

Fig. S3 - this figure indicates Ube2N~Ub is used however experiments are done in the unconjugated form

Reviewer #2 (Remarks to the Author):

In their manuscript entitled “A tri-ionic mechanism drives Ube2N-specific recruitment and K63-chain ubiquitination in TRIM ligases”, Kiss & Zeng et al. carefully describe the interactions required for TRIM21 to select the K63-specific E2 Ube2N and assemble the Ub signals necessary for virus neutralization, immune activation, and Trim-away degradation. The authors go on to expand their analysis to other RING E3 ligases in and outside of the TRIM subfamily, and reveal the tri-ionic selection mechanism to be a conserved feature across many ubiquitin ligases. The work was done to a very high standard and the data are presented beautifully through both well-written text and wonderful figures. The conclusions will be appreciated by researchers from numerous fields, as the tri-ionic motif appears to be highly conserved and provides new tools for testing ligase function in other systems. As the authors highlight, this work also addresses a very perplexing area of ubiquitin research, i.e. how are E3:E2 pairs regulated in cells. I highly recommend publication with only minor comments detailed below.

- 1) Please dial back or clarify the statements of novelty regarding the first example of E2-specific recruitment. As the authors point out, this has also been shown for FANCL:Ube2T, but also for Dcn1:Ubc12 (Scott et al., 2011) and, to a lesser extent, other examples as well. The tri-ionic motif does, however, appear to be the most generalizable mechanism of E3:E2 selectivity identified to-date.
- 2) Line 132: I believe this figure reference could be Fig. 1d-f inclusive.
- 3) Figure 2c-e: Labeling TRIM21 residues differently for proximal and distal moieties would be helpful, e.g. E13p and R67d.
- 4) Line 274: I believe there should be a reference to Fig. 6c here, otherwise I don't think it is referenced.

- 5) Supplementary Fig. 5 could use more labeling or explanation in the legend to help the reader understand the data.
- 6) Line 380: I believe you intended to refer to Supplementary Fig. 8 here? Please also provide an in-text reference for your statement regarding cellular E2 concentrations.

Sincerely,

Jonathan Pruneda

Reviewer #3 (Remarks to the Author):

In this manuscript, Kiss et al report structural, biochemical, and cell biological studies showing how the TRIM21 RING domain engages the E2 conjugating enzyme Ube2N to generate K63-linked polyubiquitin chains, previously shown to underlie antibody-dependent antiviral mechanisms and a protein depletion method called TRIM-away. In a series of studies, the authors show that: (1) TRIM21 RING domain can catalyze ubiquitination in vitro with a redundant series of E2 enzymes, including Ube2W and Ube2N (as previously shown). Three Ube2D enzymes also show significant activity in vitro, both with regards to binding and ubiquitination. However, in cells depleted of these E2s antibody-dependent virus neutralization is not impaired, indicating that TRIM21 can differentiate between different E2s in cells in a manner not readily observable in vitro. (2) The authors then solved a crystal structure of the TRIM21 RING domain in complex with a stable Ube2N-Ub conjugate. Many of details of this structure recapitulate various known structures of dimeric RING domains in complex with E2s or E2-Ub conjugates. The authors focus on a “tri-ionic motif” consisting of E12, E13 and D21 that are described to provide “ideally distributed anchor points” to capture Ube2N-Ub in its active, so-called “closed” conformation. (3) Authors then test the E12-E13 di-glu as anchor point for binding of Ube2N, by performing NMR chemical shift perturbation experiments against labeled monomeric TRIM21 RING (having the M10E mutation). This series of experiments support roles for E13/E13 in binding, but effects were modest. Comparing results with titration of Ube2N-Ub, effects due to ubiquitin were observed for a number of residues (E12, V53, V54, R55), in agreement with proximal RING/Ub interactions observed in the crystal structure. E12 and E143 mutations led to a reduction in ubiquitin-specific signals, interpreted to mean that anchor point residues enable catalysis by binding to the closed Ub conformation. (4) Authors then use ubiquitin discharge assays to test impact of “anchor point mutants” and found abrogated discharge with Ube2N but not Ube2D. This biochemistry section is crucial to their primary model of E2:E3 specificity. (5) Mutants were tested in cells for ability to support virus neutralization and TRIM-away, with various degrees of impairment observed. (6) Authors then show conservation of anchor motif in TRIM5, another antiviral TRIM, and in other RINGs.

The major claim here is that the authors have uncovered a tri-ionic anchor motif as a major mechanism of E2:E3 specificity in RING domains. Very nice model, with nice supporting data. However, the claim is so strong that the authors should demonstrate the predictive power of their model, by switching the specificities of Ube2N and Ube2D. Otherwise, they need to soften their interpretation.

Major comments:

(1) One major difficulty in reading and reviewing this manuscript is that the authors do not attempt to distinguish the structural features observed in their crystal structure (salt bridges, hydrogen bonds, etc.) from their interpretation of the functional significance (ideal anchor points, stabilizing closed conformation, specific recruitment). This style of writing gave this reviewer a strong impression that the model was being prioritized over data and made it very difficult to parse out relative strengths of evidence being presented.

(2) Data in Fig. 4 are critical for their argument of E2:E3 specificity, and are not that strong. If true that E2:E3 salt bridges involving anchor motifs are the sole determinants of specificity, then they should be able to switch specificities of Ube2N and Ube2D. Otherwise, the claim needs to be softened significantly.

(3) All other supporting data seem adequate, but collectively do not quite reach the level of evidence needed to support the model.

Other:

Fig. 2: It would be helpful to box out in panel a the regions shown in panels b-e.

Fig. 4a,b,f: No error bars; not stated how many replicates (technical or biological).

Line 155: "ideally distributed anchor points" – not clear why these are ideal.

Line 161: "in contrast to the general mechanism of E2:E3 interaction" – clarify what is being contrasted here; perhaps in reference to "generic interactions" described in previous paragraph?

Lines 177-179: "The di-glutamate motif forms salt bridges ..." – this sentence refers to a structural feature observed in the crystal structure. It needs to be re-written in order to avoid the impression that salt-bridge formation has been also inferred from the NMR titration data. Throughout the manuscript, E2:E3 interactions are assumed to involve RING E3s; this should be explicitly stated early in the ms to avoid confusing casual readers because there are other types of E3s.

In particular, it will be important to perform the additional experiments and analyses suggested by Reviewer 1, and to fully address Reviewer 3's concerns

We are grateful for the reviewer's time and work on our manuscript. We have addressed all concerns and performed additional biochemical and biophysical experiments, which we believe have strengthened our manuscript further; we hope it is now suitable for publication in *Nature Communications*.

Reviewer 1

Firstly, the E2 screen in Fig. 1a and Fig. S1 is rather ambiguous. The authors claim that Ube2D1-4 function with T21-R yet the blots in Fig. S1 do not show this. There is a large unresolved section of the Ub-IB that might be attributed to significant ubiquitination but this does not show up on the lower gels. In addition, one would expect that some discrete bands would be visible on the IB as they are for Ube2N/V1 and Ube2N/V2. Further, if the large blot is positive ubiquitination as the authors suggest, one would conclude that Ube2D1-4 are the most active E2s. This would not be consistent with the authors conclusions from the crystal structures. Further, if activity can not be shown one would question the use of Ube2D1 in other experiments. This assay needs to be repeated and clarified.

We thank the reviewer for their detailed examination of the E2 screen and agree that the ubiquitination with Ube2D1-4 looks unexpectedly different from the laddering observed with Ube2N. To confirm that the observed high molecular weight products are indeed ubiquitin chains, we performed a ubiquitination time course experiment with Ube2D1 and T21-R, which we have added as new data in Supplementary Fig. 1d. This experiment shows that with T21-R, Ube2D1 only produces very long ubiquitin chains, as we also observed in our biochemical experiments in Fig. 4g. After 5 minutes incubation, we resolve very long ubiquitin chains that become longer over the course of the experiment. This behaviour is in agreement with previous publications from the Huang lab., who showed that backside ubiquitin binding to Ube2D2 promotes activity of the E2, which greatly favours formation of longer ubiquitin chains (Buetow et al., 2015, Mol Cell; Patel et al., 2018, J Biol Chem).

A direct activity comparison between E2s is complicated by differences in their extension mechanisms and ubiquitin chain products. We have also limited our interpretation of the screen data due to the nature of the assay. The biochemical screen used in Supplementary Fig. 1a-c is a commercial kit, purchased from Boston Biochem (E2Select Ubiquitin Conjugating Kit (K-982, Boston Biochem, Cambridge, USA)), in which the concentrations of the different E2 enzymes are not the same. Attempting to normalise this is problematic as cellular concentrations of Ube2D and Ube2N are very different (Supplementary Fig. 13), as are the number of E3's that can interact with these E2 enzymes and their competition for E2 binding. We have therefore used the screen purely as a means of detecting the presence or absence of biochemical activity with TRIM21. Going beyond the screen, one of the findings of our study is that in vitro E2 activity doesn't necessarily match cell phenotypes. Despite the high activity of Ube2D with TRIM21 in vitro, we show that Ube2D is not required for TRIM21's cellular antiviral function (Fig. 1d).

From the NMR data it is clear that Ube2N (Ube2D1) bind quite weakly to T21-R. From the crystal structure one would predict that binding should be enhanced in the E2~Ub conjugate. This should be quantified.

We are grateful for this suggestion as an analysis of the binding affinity provides an opportunity to complement our structural data on E2~Ub interaction. As our NMR titrations were performed with monomeric T21-R^{M10E}, we carried out a new set of experiments using an alternative technique, surface plasmon resonance (SPR), to allow measurement of the wild-type RING. We measured a K_D of 43 μ M for the binding of Ube2N and 5 μ M for the binding of Ube2N~Ub (Fig. 3a,b). This enhancement in affinity of approximately one order of magnitude is consistent with additional contacts being provided by ubiquitin with the distal and proximal RING in the dimer (see Fig. 2). This result is also in agreement with published data showing that RING E3 ligases have a higher affinity for Ube2D2~Ub compared to unconjugated Ube2D2 (Duo et al., 2012, NSMB; Buetow et al., 2015, Mol Cell). We have added this new data to Fig. 3 as panel a. The measured K_D between GST-T21-R and Ube2N is 2.8-times higher than the 15 μ M measured by NMR between T21-R^{M10E} and Ube2N (Supplementary Fig. 3c). This slight difference comes most likely from the different experimental conditions (pH, concentrations).

Despite the importance of E12, E13 in T21-R the CSPs in this region are minimal. This should be discussed.

We thank the reviewer for carefully analysing our data. We assume this is in reference to the NMR titrations of T21-R^{M10E} with Ube2N, shown in Supplementary Fig. 3. While it is of course true that CSP titrations are very commonly used (including in this work) to detect structural perturbations, there is no strictly quantitative link between the size of individual CSPs and the extent of nearby atomic movements or the strength of individual binding interactions. In the case of amide group CSPs, as measured here, a particularly strong influence comes from any changes in hydrogen bonding. This can for instance be seen from the strong CSP that we observe for residues I18 and L20 when Ube2N is added (Supplementary Fig. 3a); the main chains of these two residues engage in hydrogen bonds with Ube2N, as can be observed in the crystal structure (Fig. 2c). In contrast, E12 and E13 lie within an alpha-helix. Both in the free RING domain structure (see 5OLM, Dickson et al., 2018, eLife) and in the complex (Fig. 2), their local hydrogen bond network remains unchanged, which we believe explains the low CSPs observed for these amides, even though the distal ends of the sidechains of these residues engage Ube2N/Ubiquitin via salt bridges. We have modified the main text discussing NMR data to clarify these points.

The alignment in 4E should be applied to all E2s tested in Fig. 1 to show that none of the other E2 enzymes (ie. non-functional) fulfill the basic residue requirement.

As requested, we have performed a clustal omega alignment of all E2s present in the screen in Fig. 1a/Supplementary Fig. 1 and added this as Supplementary Fig. 9. In addition to Ube2N, the requirement for suitably placed basic residues might potentially be fulfilled in the non-canonical E2 enzyme Ube2J2. However, Ube2J2 is a membrane bound E2 that is involved in ER quality control and shows a serine/threonine specificity (Steward et al., 2016,

Cell Res; Ye and Rape, 2009, Nat Rev Mol Cell Biol). Bakers yeast Ube2J2 (scUbc6) was shown to interact with the membrane-bound E3-ligase Doa10 and to act as a priming E2 enzyme. It adds monoubiquitin to the substrate, which is then extended by a second, different E2 enzyme (scUbc7) (Weber et al., 2016, Mol Cell). It would be interesting to understand the role of the basic residues in this system and which of the many differences between Ube2J2 and Ube2N are responsible for lack of TRIM21 engagement but this is beyond the scope of the current study.

On p8.9 the authors describe experiments to confirm the closed E2~Ub conformation in solution by monitoring CSPs in T21-R with Ube2N and Ube2N-Ub (Fig. 3). Although different CSPs can be noted for these two titrations this data does not show the closed conformation but rather CSPs from E2 or E2-Ub binding. In order to monitor the closed conformation the authors would need to monitor the Ub or E2 within the conjugate as has been done by several other groups. This is a straight-forward experiment.

We thank the reviewer for the careful analysis of our experiment and agree that although our results strongly suggest that the dCSPs indicate the formation of the closed conformation, this could be verified by collecting additional data. We have therefore performed the suggested experiment and isotopically labelled Ub and conjugated it to Ube2N. We then performed triple resonance assignment experiments with a Ube2N~¹⁵N/¹³C-Ub construct to assign the ubiquitin peaks when conjugated to the E2 (Supplementary Fig. 6). Even in the absence of an E3 ligase, we observed CSPs that indicate partial occupancy of a closed conformation (Supplementary Fig. 5a-c), in agreement with previous studies (Pruneda et al., 2011, Biochemistry). Next, we titrated T21-R^{M10E} into Ube2N~¹⁵N-Ub (Supplementary Figs. 5d-e). The CSPs we obtained agree well with a further conformational shift towards the closed Ube2N~Ub conformation when in presence of T21-R^{M10E}. This new data is discussed in the third part of the results section.

Fig. S3 - this figure indicates Ube2N~Ub is used however experiments are done in the unconjugated form

We thank the reviewer for informing us of this error; we have changed the Figure title accordingly.

Reviewer 2

Minor comments:

Please dial back or clarify the statements of novelty regarding the first example of E2-specific recruitment. As the authors point out, this has also been shown for FANCL:Ube2T, but also for Dcn1:Ubc12 (Scott et al., 2011) and, to a lesser extent, other examples as well. The tri-ionic motif does, however, appear to be the most generalizable mechanism of E3:E2 selectivity identified to-date.

We thank Prof. Pruneda for drawing our attention to the excellent Scott et al. (2011, Science) paper. This paper, together with subsequent work (Scott et al., 2014, Cell), describes how cullins are neddylated. During this process, Ubc12 (Nedd8 E2 enzyme) forms

a closed conformation with the cullin RING (Rbx1). In addition, the N-terminal helix of Ubc12 is acetylated and bound by the protein Dcn1, which accelerates Neddylation. The acetylation of the Ubc12 N-terminus indeed makes the interaction with Dcn1 specific. This mechanism is arguably important more for explaining how the E2 is activated rather than its E3 specificity, analogously to Ube2N that is activated by binding of its partner, the inactive Ube2V2 (Branigan et al., 2015, NSMB). In our revised manuscript we now refer to this Dcn1:Ubc12 mechanism in the discussion, immediately after mention of FANCL:Ube2T. We have also sought to properly constrain our statements of novelty in the text. For instance in the abstract: "...is the first example of a general E2-specific catalysis mechanism of RING E3s in cells".

Line 132: I believe this figure reference could be Fig. 1d-f inclusive.

We thank Prof. Pruneda for spotting this error. We have corrected the Figure reference.

Figure 2c-e: Labeling TRIM21 residues differently for proximal and distal moieties would be helpful, e.g. E13p and R67d.

We thank Prof. Pruneda for suggesting this. We have labelled the distal and proximal RINGS and their labels in different shades of blue.

Line 274: I believe there should be a reference to Fig. 6c here, otherwise I don't think it is referenced.

We have edited the Figure reference accordingly.

Supplementary Fig. 5 could use more labeling or explanation in the legend to help the reader understand the data.

As suggested, we have clarified this Figure and modified the Figure legend.

Line 380: I believe you intended to refer to Supplementary Fig. 8 here? Please also provide an in-text reference for your statement regarding cellular E2 concentrations.

The Figure reference and citations were accidentally deleted during submission of the manuscript. We have added them back into the text.

Reviewer 3

Major comments:

One major difficulty in reading and reviewing this manuscript is that the authors do not attempt to distinguish the structural features observed in their crystal structure (salt bridges, hydrogen bonds, etc.) from their interpretation of the functional significance (ideal anchor points, stabilizing closed conformation, specific recruitment). This style of writing gave this reviewer a strong impression that the model was being prioritized over data and made it very difficult to parse out relative strengths of evidence being presented.

We apologise that by focussing on linking data from the different types of experiment carried out in the study may have unintentionally obscured the primary results from our interpretations. To correct this, we have edited the manuscript to first describe the result then compare it to the hypothesised mechanism. For example, when presenting the crystal structure (results: Crystal structure of TRIM21-RING with Ube2N~Ub; page 6, line 138 until page 8, line 170) we now first list all features of the structure (salt bridges, hydrogen bonds, etc.), and then move on to why they might be important (e.g. ideal anchor points). Moreover, we have altered the style of wording in the results sections, for instance “Mutation of the third anchor point to arginine (D21R)...” to a more factual style (e.g. “Mutation of D21 to arginine ...” page 10, line 233).

Data in Fig. 4 are critical for their argument of E2:E3 specificity, and are not that strong. If true that E2:E3 salt bridges involving anchor motifs are the sole determinants of specificity, then they should be able to switch specificities of Ube2N and Ube2D. Otherwise, the claim needs to be softened significantly.

We agree with the reviewer that the data in Fig. 4 is critical to our argument of E2 specificity. With the combination of structural biology, biophysics, biochemistry and cell biology that we have used, we are confident that the tri-ionic motif is critical for Ube2N recruitment and catalysis. As suggested by the reviewer, we attempted to switch specificities by incorporating the arginine from Ube2N into Ube2D and the aspartate from Ube2D into Ube2N. In Figure 4h,i, Ube2N^{R14D} swaps the arginine that naturally interacts with TRIM21^{E12} to the acid in Ube2D1 and Ube2D1^{D12R} vice versa. In our ubiquitination assay, Ube2N^{R14D} lost most of its activity with T21-R (Fig. 4h). In contrast, Ube2D1^{D12R} remained active with T21-R (Fig. 4i). Loss of Ube2N^{R14D} activity is consistent with the lack of catalysis by the corresponding T21-R^{E12R} mutation and supports the importance of the salt bridge. Meanwhile, the lack of change in Ube2D1^{D12R} is consistent with activity involving this E2 being independent of the salt bridge.

We also tested whether the corresponding charge swap in T21-R^{E12R} would alter the activity of the mutant E2s. As expected, T21-R^{E12R} was active with both Ube2D1 and Ube2D1^{D12R} enzymes (Fig. 4i), as Ube2D interaction is not dependent on the tri-ionic motif. Switching the specificity of Ube2D and making it Ube2N-like would first require removing enough of the promiscuous interactions to allow a gain-of-function to be observed. In contrast, a salt bridge equivalent to that between E12(T21) and R14(Ube2N) could be restored by using T21-R^{E12R} with Ube2N^{R14D}, leading to a potential increase in activity. However, in our ubiquitination assay, Ube2N^{R14D} remained largely inactive with T21-R^{E12R} (Fig. 4h). This apparent difference between theory and experiment is likely because the residues surrounding position 14 may influence, and be influenced by, switching the charge. Coulomb forces can act over long distances (e.g. the pK_a of subtilisin can be significantly changes by alterations ~15 Å from the catalytic base; Russel & Fersht, 1987, Nature). Swapping R14 on Ube2N will influence the pK_a of nearby residues important for TRIM21 interaction (e.g. K10 is 6.1 Å and R6 is 12.3 Å away from R14). In addition, while T21-R^{E12R} is still an active E3 (Fig. 4f,g,i), we do not know whether the R14D mutation has simply killed Ube2N activity (e.g. due to reasons of pK_a given above).

Although we agree with the referee that engineering of Ube2N-selectivity based on the tri-ionic motif would be highly interesting, our preliminary experiments during this revision have shown that this is not a simple task and we believe achieving it would not be within the scope of the present manuscript. However, this is a question we are interested in tackling in the future as part of a more comprehensive protein engineering and enzyme evolution project.

All other supporting data seem adequate, but collectively do not quite reach the level of evidence needed to support the model.

With the additional experiments we have performed during our revision we believe that our data – crystallographic, NMR, SPR, enzymatic and cellular neutralization, signalling and Trim-Away assays – all support our proposed model that a tri-ionic motif is critical for TRIM21 catalysis of Ube2N ubiquitination and cell function. The crystallographic data reveal the three salt bridges formed by the tri-ionic motif (Fig. 2). Characterization of the contribution of these residues to E2 interaction by NMR demonstrate that they are essential for formation of the closed (enzymatically active) conformation (Fig. 3). Enzymatic assays comparing mutants of the tri-ionic motif confirm that this is critical for catalytic function with Ube2N (Fig. 4). Matched experiments with Ube2D1 demonstrate that requirement for the motif is E2-specific and that Ube2N and Ube2D1 are fundamentally different in the manner in which they engage the RING E3 (Fig. 4). Finally, we show that the tri-ionic motif mechanism of catalysis is essential for multiple TRIM21 cell phenotypes (Figs. 1 and 5). Because mutants of the tri-ionic motif specifically impair catalysis with Ube2N, but not Ube2D, this has led to the discovery that TRIM21-catalysis of K63-linked ubiquitin chains is absolutely critical for antibody-dependent virus neutralization, immune activation and protein depletion by Trim-Away (Fig. 5). Moreover, the data on the selective importance of the tri-ionic motif for Ube2N activity is consistent with our finding that Ube2D is not required for TRIM21 cellular function (Fig. 1). In summary, we have approached the question of Ube2N selectivity using a wide range of diverse methods that all support a clear molecular model, such that we are confident to propose the tri-ionic motif as a key mechanism providing Ube2N-specificity for TRIM-RINGs.

Minor comments:

Fig. 2: It would be helpful to box out in panel a the regions shown in panels b-e.

We thank the reviewer for this suggestion and have added boxes to make clear where the close-up Figures come from.

Fig. 4a,b,f: No error bars; not stated how many replicates (technical or biological).

Biochemical assays were performed in at least two independent experiments. The E2-discharge assays in Fig. 4a,b,f have no error bars as they are a representative single experiment and are quantification of the bands in the neighbouring blot. Cell biological data is from at least three independent experiments and the graphs show the average of these biological replicates (mean \pm SEM). We have clarified this in the figure legend.

Line 155: “ideally distributed anchor points” – not clear why these are ideal.

We interpreted the residues of the tri-ionic motif as ideally distributed anchor points, because they have precisely the correct position to allow them to specifically interact with both ends of the first helix of Ube2N and simultaneously bind ubiquitin. Together, they wrap the Ube2N~Ub around the RING domain and lock it in the closed conformation.

Line 161: “in contrast to the general mechanism of E2:E3 interaction” – clarify what is being contrasted here; perhaps in reference to “generic interactions” described in previous paragraph?

We thank the reviewer for suggesting clarification of this point. Indeed, with general mechanism we were referring to the generic interactions described in the previous paragraph. We have changed this sentence accordingly.

Lines 177-179: “The di-glutamate motif forms salt bridges ...” – this sentence refers to a structural feature observed in the crystal structure. It needs to be re-written in order to avoid the impression that salt-bridge formation has been also inferred from the NMR titration data. Throughout the manuscript, E2:E3 interactions are assumed to involve RING E3s; this should be explicitly stated early in the ms to avoid confusing casual readers because there are other types of E3s.

We thank the reviewer for suggesting this clarification. We have fixed the sentence accordingly by making clear that we know this from the crystal structure. It is true that the ~600 E3 ligases also include 28 HECT and 12 RBR type ligases. However, to ensure that there is no confusion, we only mention the RING type in the manuscript, starting from the introduction.

REVIEWERS' COMMENTS:

Reviewer #1 (Remarks to the Author):

This authors have addressed my previous comments. The inclusion of new data and clarification significantly improves this manuscript.

Reviewer #3 (Remarks to the Author):

The reviewers have thoroughly addressed my critique. I have no further comments.